# Rethinking Nighttime Image Deraining via Learnable Color Space Transformation

**Qiyuan Guan**[*]
Dalian Polytechnic University
qyuanguan@gmail.com

**Xiang Chen**[*]
Nanjing University of
Science and Technology
chenxiang@njust.edu.cn

**Guiyue Jin**[†]
Dalian Polytechnic University
guiyue.jin@dlpu.edu.cn

**Jiyu Jin**[†]
Dalian Polytechnic University
jiyu.jin@dlpu.edu.cn

**Shumin Fan**
Dalian Martime University
shuminfan@163.com

**Tianyu Song**
Dalian Martime University
songtienyu@163.com

**Jinshan Pan**
Nanjing University of Science and Technology
jspan@njust.edu.cn

## Abstract

Compared to daytime image deraining, nighttime image deraining poses significant challenges due to inherent complexities of nighttime scenarios and the lack of high-quality datasets that accurately represent the coupling effect between rain and illumination. In this paper, we rethink the task of nighttime image deraining and contribute a new high-quality benchmark, HQ-NightRain, which offers higher harmony and realism compared to existing datasets. In addition, we develop an effective Color Space Transformation Network (CST-Net) for better removing complex rain from nighttime scenes. Specifically, we propose a learnable color space converter (CSC) to better facilitate rain removal in the Y channel, as nighttime rain is more pronounced in the Y channel compared to the RGB color space. To capture illumination information for guiding nighttime deraining, implicit illumination guidance is introduced enabling the learned features to improve the model's robustness in complex scenarios. Extensive experiments show the value of our dataset and the effectiveness of our method. The source code and datasets are available at *https://github.com/guanqiyuan/CST-Net*.

## 1   Introduction

Images degraded by complex nighttime rain significantly affect downstream vision tasks, such as autonomous driving and video surveillance [8, 35, 58]. Unlike daytime images, nighttime scenes present unique challenges due to low illumination and varying light sources, which together amplify the visibility and complexity of rain artifacts. Thus, it is of great interest to develop an effective algorithm to recover high-quality rain-free images from nighttime scenarios. Recently, several efforts [35, 22, 7, 24, 16] have been made to address nighttime image deraining, with the emergence of some synthetic datasets, including GTAV-NightRain [53] and Raindrop Clarity [30]. However, we note that most existing synthetic datasets [2, 46, 21] feature a global uniform distribution of rain, as random rain masks are added linearly to nighttime backgrounds. This naturally leads to

---

[*]Co-first authorship

[†]Corresponding author

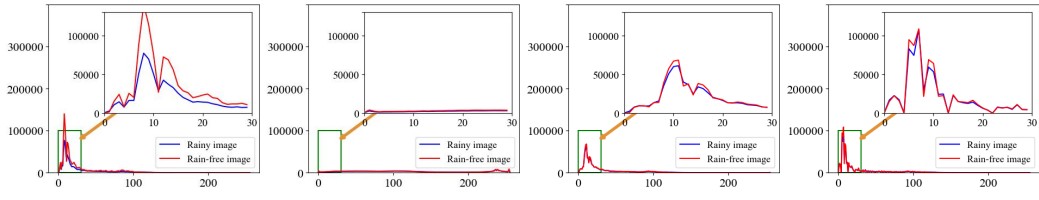

|                           |                          |                           |                           |
|:-------------------------:|:------------------------:|:-------------------------:|:-------------------------:|
| (a) Y (YCbCr); Nighttime  | (b) Y (YCbCr); Daytime   | (c) V (HSV); Nighttime    | (d) L (LAB); Nighttime    |

Figure 1: Histograms of different color channels show that the Y channel in the YCbCr color space demonstrates the most significant difference between rainy and rain-free images at nighttime.

inadequate realism and harmony in the visual appearance, which limits the ability to train better nighttime deraining models.

In fact, nighttime scenarios lack uniform global illumination, with darkness and irregular artificial light sources dominating the visual landscape [18, 31, 56]. This results in markedly different appearances and visibility distributions of rain effects, which are typically concentrated around light sources and only visible under specific lighting conditions.

Based on these observations, we design a new nighttime rainy image composition pipeline, and contribute a high-quality benchmark dataset, HQ-NightRain. In our pipeline, we fully consider the visibility of rain under varying illumination conditions. Specifically, our approach involves associating the rain mask with an illumination coefficient matrix of the nighttime background image, thereby generating a non-uniform nighttime rain distribution to more realistically adapt nighttime scenes. By this way, we ensure the nighttime rain is visually more compatible with the corresponding background scenes, aiming at reducing the domain gap between synthetic and real-world datasets.

Armed with this dataset, our focus shifts to exploring an effective algorithm for nighttime image deraining. We note that existing nighttime deraining approaches [35, 54] do not take into account the inherent properties of nighttime rainy images. In other words, these methods still perform image deraining in the RGB color space. Based on the pixel value statistical analysis in Figure 1, we find that nighttime rain is most pronounced in the Y channel. The reason behind this lies in the fact that the Y channel in the YCbCr color space captures luminance information. In low-light conditions, rain often reflects light from artificial sources, creating high-contrast patterns that stand out sharply against the dark background. This makes the differences between rainy and rain-free images more prominent in the Y channel compared to the RGB channels. Therefore, this motivates us to develop an effective approach performed in the Y channel, tailored for nighttime image deraining.

To this end, we propose an effective color space transformation framework CST-Net for nighttime image deraining. Specifically, we develop a learnable color space converter that transforms the input from the RGB space to the YCbCr space to perform rain degradation removal. This enables the model to adaptively allocate the parameters required for color space transformation across various nighttime images, resulting in better robustness in real-world scenarios. To better represent illumination information, we introduce implicit illumination guidance to enhance nighttime rain removal. Extensive experiments demonstrate that our method achieves favorable performance against state-of-the-art ones on our proposed dataset and public benchmarks. The main contributions are summarized as follows:

- We construct a high-quality nighttime image deraining benchmark dataset HQ-NightRain, which further improves the harmony and realism of synthetic images.

- We propose a robust learnable color space transformation framework for nighttime image deraining, which explores the potential of the Y channel in rain removal.

- We demonstrate the effectiveness of our dataset, and show that our proposed method achieves favorable deraining performance against state-of-the-art ones.

## 2   Related Work

**Image deraining**. Recent years have witnessed significant advancements in image deraining, driven by the emergence of numerous benchmark datasets and deep learning models [7, 25, 26, 41]. We note

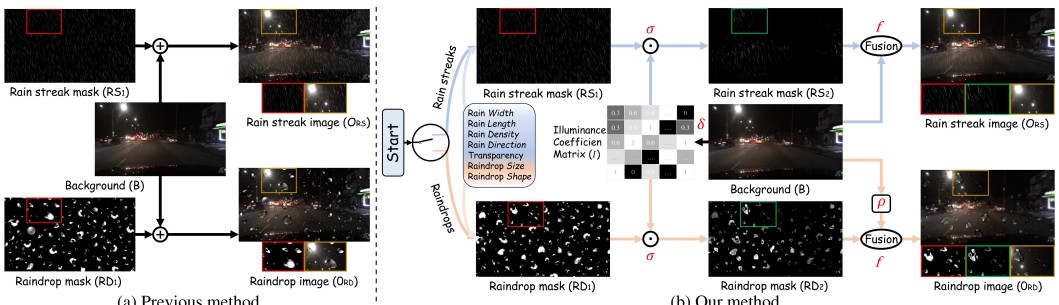

Figure 2: Data construction pipeline of previous method and our proposed approach. Existing methods add global rain effects linearly onto nighttime backgrounds to generate rainy images. Insetad, our approach takes illumination information in nighttime scenes into account to synthesize nighttime rainy images with higher harmony and realism.

existing image deraining efforts primarily focus on daytime scenarios [47], with comparatively little attention given to nighttime deraining. For the **nighttime deraining dataset**, Zhang *et al.* [53] first proposed the GTAV-NightRain dataset for nighttime rain streak removal, using ray tracing technology on a gaming platform to render rainy scenes. Li *et al.* [35] synthesized the RoadScene dataset using a GAN-based method, specifically designed for nighttime driving scenarios in rainy conditions. Recently, Jin *et al.* [22] constructed a dual-focused dataset for day and night raindrop removal. Although these datasets have been proposed, most synthesis methods rely on approaches used for daytime rainy scenes, resulting in a large domain gap between synthetic and real images. Instead, we build a high-quality benchmark by taking into account illumination properties of nighttime images.

For the **nighttime deraining method**, Zhang *et al.* [54] proposed a rain location prior (RLP) that employs a recurrent residual network to learn the positional information of rain streaks from nighttime rainy images. Lin *et al.* [30] developed a teacher-student framework with an adaptive deraining module and an adaptive correction module to remove rain from nighttime videos. Although these methods have made preliminary explorations in nighttime deraining, they overlook the illumination properties of nighttime rain and continue to perform deraining in the RGB space, similar to daytime deraining approaches [6, 4]. Different from these methods, we leverage a learnable color space transformation to perform rain removal in the YCbCr color space.

**Color space transformation**. Color space transformation refers to the process of converting an image from one color space to another to facilitate various image processing tasks. As a result, color spaces like YCbCr, HSV, HVI, *etc.*, have been explored in recent years [23, 38, 43]. For example, the RGB color space uses red, green, and blue components, while the YCbCr space separates luminance (Y) from chrominance (Cb and Cr). In nighttime rainy images, rain is often more pronounced in the Y channel compared to the RGB channels. Nighttime rain can significantly affect the overall brightness, making them more visible in the Y channel [17]. In this work, by focusing on the Y channel, we can better distinguish between rain and background, thus enhancing the effectiveness of nighttime deraining models.

## 3 Dataset Construction: HQ-NightRain

Existing nighttime deraining datasets [35, 22, 53] often lack realism and visual harmony due to the linear addition of rain effects onto nighttime backgrounds. In this paper, we develop an illumination-fused image synthesis method to generate more realistic nighttime rainy images.

### 3.1 Nighttime Rain Model

Most previous studies [21, 35, 22] use a linear superposition model to synthesize rain streak images $R_s$ and raindrop images $R_d$. Mathematically, it can be expressed as:

$$R_s = B + S, \quad R_d = (1 - M) \odot B + D, \tag{1}$$

where $B$ represents the rain-free nighttime background, $S$ is the rain streak mask, $D$ is the raindrop mask, $M$ is a binary mask, and $\odot$ denotes element-wise multiplication.

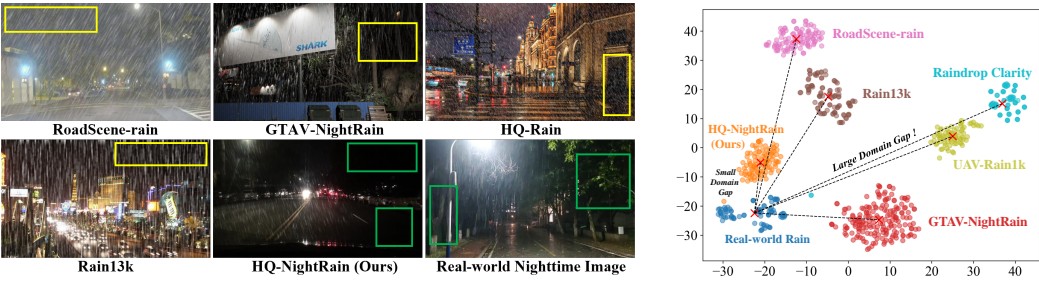

(a) Comparison of different nighttime rainy visual samples      (b) Distributions of t-SNE

Figure 3: (a) Sample images from different datasets. Our dataset carefully considers harmony in synthesizing nighttime rainy scenes, resulting in more visually realistic appearances. (b) To avoid the influence of background content on the results, we select samples with similar nighttime street backgrounds and use ResNet50 to extract features. Images from the same dataset exhibit clustering after t-SNE dimensionality reduction due to the similarity of rain streak features. The small domain gap indicates that our dataset synthesis pipeline generates nighttime rain that is closer to reality.

In this work, we rethink the formation of nighttime rainy images by examining the visibility and distribution characteristics of nighttime rain effect [31]. During the day, light is produced by the sun, which can be considered parallel and uniform, making rain clearly visible. At night, the primary light sources are artificial (*e.g.*, street lamps, traffic lights, vehicle headlights), and the light radiates outwards from these sources. According to the first law of illumination [9], the illuminance on a surface is inversely proportional to the square of the distance from the light source. Therefore, rain in nighttime scenes is only visible near light sources and is not noticeable in low-illumination areas.

Based on this observation, we propose a new rain model for nighttime rain imaging, defined as:

$$R_s = f[B, \sigma(S)], \quad R_d = f[(1 - M) \odot \rho(B), \sigma(D)], \tag{2}$$

where the function $\sigma(\cdot)$ incorporates illumination information into the rain streak and raindrop masks, $f[\cdot]$ represents the 3×3 convolution function used to merge the background with rain mask, and $\rho(\cdot)$ denotes the defocus blur operation.

## 3.2 Data Construction Pipeline

We present the data construction pipeline of HQ-NightRain in Figure 2. Compared to existing nighttime rainy datasets that only consider uniform rain distribution, our approach integrates lighting information from nighttime scenes, allowing the appearance of rain streaks and raindrops to be non-uniformly distributed based on illumination, which is commonly observed in real-world scenarios.

**Illumination coefficient estimation**. To fully account for the characteristics of nighttime images, we first extract the illumination coefficient matrix from the background image. Specifically, we convert the background image $B$ from the RGB color space to the HSV color space. Then, we extract illumination information from the V channel of the background image. This process is defined as:

$$\mathbf{N} = Norm\left(f_v\left(\mathcal{T}_{RGB2HSV}\left(B\right)\right)\right), \tag{3}$$

where $\mathbf{N}$ denotes the initial illumination matrix, $\mathcal{T}$ denotes the color space transformation, $f_v$ denotes the extraction of the V channel, and $Norm$ is the normalization function.

Rain is not only invisible in low-illumination areas but also in excessively high-illumination areas, such as at the center of a light source. To this end, we apply a masking operation $Mask(\cdot)$ to $\mathbf{N}$, setting high and low illumination thresholds to identify the illuminated regions where nighttime rain is visible. The final illumination coefficient matrix $\mathbf{I}$ is expressed as follows:

$$\mathbf{I} = Mask(\mathbf{N}) = \begin{cases} v/2, & \text{if}(\mathbf{N}_{(x,y)} \leqslant \tau_1 \parallel \mathbf{N}_{(x,y)} \geqslant \tau_2) \\ v, & \text{other} \end{cases}, \tag{4}$$

where $v$ is the value at position $(x, y)$ in matrix $\mathbf{N}$, $\tau_1$ and $\tau_2$ are denote the illumination thresholds.

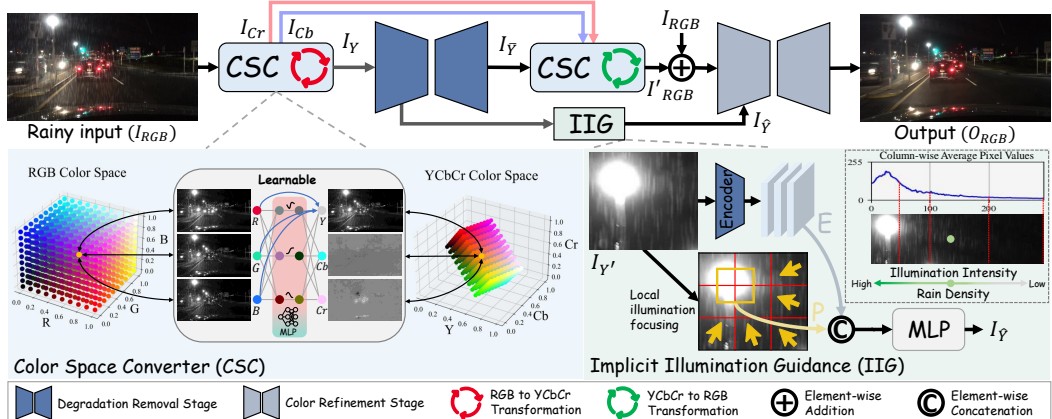

Figure 4: Overall architecture of the proposed end-to-end CST-Net for nighttime image deraining, which consists of a degradation removal stage and a color refinement stage. Among these two stages, we develop a color space converter (CSC) to achieve space transformation (RGB $\leftrightarrows$ YCbCr), and also construct an implicit illumination guidance (IIG) branch to better transmit illumination information.

**Nighttime rain mask blending**. We first generate initial rain streak masks (with varying widths, lengths, densities, directions, and transparencies) [7] and raindrop masks (with varying sizes and shapes) [2]. Subsequently, the appearance of the obtained nighttime rain layer is associated with the illumination information of the background image, expressed as:

$$\sigma(S) = S \odot \mathbf{I}, \quad \sigma(D) = D \odot \mathbf{I}, \tag{5}$$

where $\sigma(\cdot)$ represents the blending of the illumination with the rain mask. Based on real-world observations, when light from sources such as streetlights or car headlights passes through raindrops, it produces refraction and scattering effects, resulting in a defocus blur phenomenon in nighttime rain. Therefore, we also apply the defocus blur function to the rain-free nighttime background, see Eq. (2).

### 3.3 Benchmark Statistics and Comparisons

The background images in our proposed HQ-NightRain are selected from the public BDD100K [50] dataset, which contains images captured from first-person driving scenes in urban environments. Here, we select rain-free backgrounds labeled as 'night' according to the provided JSON files. In total, our dataset contains 11,200 image pairs, with 10,000 pairs for training, 900 pairs for validation, and 300 pairs for testing. Based on the degradation type of nighttime rain, it is further divided into three subsets, including rain streak (RS), raindrop (RD), and a mixture of rain streak and raindrop (SD). Figure 3 presents comparison results between existing rainy datasets [35, 22, 53, 21] and our benchmark, demonstrating that HQ-NightRain more closely aligns with the distribution of real-world nighttime rainy images. HQ-NightRain carefully considers harmony in synthesized nighttime rainy scenes, which reduces the domain gap between synthetic and real images. We additionally provide a real-captured subset comprising 512 images and a synthetic nighttime rain subset in natural scenes comprising 20 image pairs.

## 4 Proposed Method

### 4.1 Overall Pipeline

To better model the complex rain information in nighttime images within the Y channel, we develop an effective color space transformation network CST-Net. Figure 4 summarizes the two-stage architecture of CST-Net, which consists of a degradation removal stage and a color refinement stage.

Given that the distribution of nighttime rain is more distinct in the luminance channel (*e.g.*, Y) compared to the color channels (*e.g.*, RGB), we first perform Y-channel based image deraining in the degradation removal stage. Specifically, given a rainy image $I_{RGB} \in \mathbb{R}_{RGB}^{H \times W \times 3}$, where $H$ and $W$ denote the image height and width, we implement color space transformation to facilitate the

switching between RGB and YCbCr spaces. Here, we propose a learnable Color Space Converter (CSC) that transforms the input image from the RGB space to the YCbCr space, splitting it into three channels: $I_Y \in \mathbb{R}_{YCbCr}^{H \times W \times 1}$, $I_{Cb} \in \mathbb{R}_{YCbCr}^{H \times W \times 1}$, and $I_{Cr} \in \mathbb{R}_{YCbCr}^{H \times W \times 1}$. Next, we feed $I_Y$ into the degradation removal stage to perform rain removal in the luminance channel, while the chrominance channels $I_{Cb}$ and $I_{Cr}$ are directed to the color refinement stage. This process is defined as follows:

$$I_Y, I_{Cb}, I_{Cr} = split(\text{CSC}_{RGB \rightarrow YCbCr}(I_{RGB})), I_{\bar{Y}} = \mathcal{P}_1(I_Y), \tag{6}$$

where $split$ means splitting the channels, $\mathcal{P}_1$ denotes the first stage, and $I_{\bar{Y}} \in \mathbb{R}_{YCbCr}^{H \times W \times 1}$ is the luminance channel used for degradation removal.

To better handle the complex and random rain degradation near light sources, we further introduce an Implicit Illumination Guidance (IIG) module to facilitate the rain removal process, resulting in $I_{\hat{Y}} \in \mathbb{R}_{YCbCr}^{H \times W \times 1} = \text{IIG}(I_Y)$. The output is then converted back into the RGB color space by the CSC and sented to the color refinement stage for color restoration. This process is expressed as:

$$I'_{RGB} = \text{CSC}_{YCbCr \rightarrow RGB}(cat[I_{\bar{Y}}, I_{Cb}, I_{Cr}]), \quad O_{RGB} = \mathcal{P}_2(I_{\hat{Y}}, (I_{RGB} + I'_{RGB})), \tag{7}$$

where $I'_{RGB} \in \mathbb{R}_{RGB}^{H \times W \times 3}$ represents the output from the first stage converted back to the RGB space, $cat[\cdot]$ represents element-wise concatenation, $\mathcal{P}_2$ denotes the second stage, and $O_{RGB}$ is the final derained image.

## 4.2 Learnable Color Space Converter

As a basic image processing operation, RGB to YCbCr conversion separates the brightness from the color components. This is typically achieved using fixed values $\mathbf{W}$ to perform a linear transformation, which is defined as follows:

$$\begin{bmatrix} Y \\ Cb \\ Cr \end{bmatrix} = \begin{bmatrix} 0.299 & 0.587 & 0.114 \\ -0.169 & -0.331 & 0.5 \\ 0.5 & -0.419 & -0.081 \end{bmatrix} \circ \begin{bmatrix} R \\ G \\ B \end{bmatrix}, \tag{8}$$

where $\circ$ denotes matrix multiplication.

However, this fixed color space conversion is based on a standard paradigm for general scenes. It is difficult to adapt to the complex and random nighttime rain scenarios, especially when the image's brightness is strongly influenced by artificial nighttime lighting sources [56, 11]. To this end, we develop a learnable Color Space Converter (CSC), so that the learned features are robust to complex nighttime rain degradation. Specifically, we introduce a learnable matrix $\mathbf{\Phi}$ with a shape of $3 \times 3$. Unlike the fixed-weight linear transformation in $\mathbf{W}$, our proposed CSC has no fixed linear weight matrix. Instead, each parameter is replaced by a learnable one-dimensional variable, enabling more flexibility in the transformation. This matrix is defined as follows:

$$\mathbf{\Phi} = \{\varphi_{i,j}\} = \begin{bmatrix} \varphi_{1,1} & \varphi_{1,2} & \varphi_{1,3} \\ \varphi_{2,1} & \varphi_{2,2} & \varphi_{2,3} \\ \varphi_{3,1} & \varphi_{3,2} & \varphi_{3,3} \end{bmatrix}, \tag{9}$$

where $\varphi_{i,j}$ is the learnable weights at position $(i, j)$, $i$ and $j$ denote the row and column indices.

In fact, rain streaks appear as localized brightness variations, often forming transparent or semi-transparent white lines that become more pronounced in the Y channel under low-light conditions due to reflections and refractions. Our proposed CSC enables each conversion weight parameter to be replaced by a learnable one-dimensional variable, non-linearly transformed through a multi-layer perceptron (MLP) [6]. This approach not only retains the ability to process luminance in color space conversion but also allows adaptive adjustment of parameters based on specific datasets and application scenarios. It can accommodate varying lighting conditions, scene types, and content characteristics, making the extracted features more robust to complex and random nighttime rain effects. The overall process is expressed as:

$$\begin{bmatrix} Y \\ Cb \\ Cr \end{bmatrix} = \text{MLP}(\mathbf{\Phi}) \circ \begin{bmatrix} R \\ G \\ B \end{bmatrix}, \tag{10}$$

where $\text{MLP}(\cdot)$ denotes the operation performed by the MLP.

Table 1: Quantitative evaluations on the HQ-NightRain dataset and GTAV-NightRain [53] dataset. The best and second-best values are **blod** and underlined.

| Datasets | HQ-NightRain | | | | | | | | | GTAV-NightRain | | | Average | | |
|---|---|---|---|---|---|---|---|---|---|---|---|---|---|---|---|
| | RS | | | RD | | | SD | | | | | | | | |
| Methods | PSNR | SSIM | LPIPS | PSNR | SSIM | LPIPS | PSNR | SSIM | LPIPS | PSNR | SSIM | LPIPS | PSNR | SSIM | LPIPS |
| PReNet | 38.5849 | 0.9842 | 0.0234 | 32.0029 | 0.9432 | 0.1709 | 34.8373 | 0.9698 | 0.0730 | 36.6332 | 0.9703 | 0.0609 | 35.5146 | 0.9669 | 0.0821 |
| RCDNet | 37.7537 | 0.9788 | 0.0357 | 31.8923 | 0.9357 | 0.1842 | 32.7796 | 0.9512 | 0.1452 | 37.0809 | 0.9703 | 0.0638 | 34.8766 | 0.9590 | 0.1072 |
| SPDNet | 40.0493 | 0.9865 | 0.0188 | 31.4772 | 0.9361 | 0.1638 | 39.0373 | 0.9825 | 0.0405 | 38.0175 | 0.9748 | 0.0462 | 37.1453 | 0.9700 | 0.0673 |
| IDT | 42.4204 | 0.9918 | 0.0116 | 33.6176 | 0.9522 | 0.1350 | 38.4873 | 0.9843 | 0.0361 | 37.5592 | 0.9744 | 0.0493 | 38.0211 | 0.9757 | 0.0580 |
| Restormer | 41.8844 | 0.9907 | 0.0145 | 33.7958 | 0.9503 | 0.1289 | 40.1790 | 0.9884 | 0.0266 | 38.1271 | 0.9772 | 0.0403 | 38.4966 | 0.9767 | 0.0526 |
| SFNet | 41.4805 | 0.9920 | 0.0139 | 33.6059 | 0.9465 | 0.1268 | 40.3011 | 0.9875 | **0.0243** | 37.5404 | 0.9738 | 0.0470 | 38.2320 | 0.9749 | 0.0530 |
| DRSformer | 42.8107 | 0.9922 | 0.0126 | 33.8452 | 0.9491 | 0.1348 | 40.4315 | **0.9886** | 0.0251 | 37.8722 | 0.9766 | 0.0415 | 38.7399 | 0.9766 | 0.0535 |
| RLP | 40.4093 | 0.9885 | 0.0167 | 29.9728 | 0.9204 | 0.1744 | 31.1297 | 0.9709 | 0.0855 | 34.9621 | 0.9600 | 0.0945 | 34.1185 | 0.9600 | 0.0928 |
| MSGNN | 27.7182 | 0.8846 | 0.2429 | 24.5151 | 0.8244 | 0.4946 | 27.6339 | 0.9078 | 0.2884 | 34.7993 | 0.9562 | 0.1180 | 28.6666 | 0.8933 | 0.2860 |
| NeRD-Rain | 42.7139 | 0.9923 | 0.0109 | 33.8313 | 0.9500 | 0.1391 | 39.6834 | 0.9855 | 0.0320 | 37.8137 | 0.9738 | 0.0530 | 38.5106 | 0.9754 | 0.0588 |
| CST-Net (Ours) | **42.8850** | **0.9924** | **0.0100** | **33.9395** | **0.9523** | **0.1239** | **40.4984** | 0.9881 | 0.0248 | **38.9378** | **0.9786** | **0.0320** | **39.0652** | **0.9778** | **0.0477** |

Table 2: Quantitative evaluations on the RealRain-1k [28] dataset and RainDS-real [33] dataset.

| Datasets | RealRain-1k | | | | | | RainDS-real | | | | | | | | |
|---|---|---|---|---|---|---|---|---|---|---|---|---|---|---|---|
| | RealRain-1k-L | | | RealRain-1k-H | | | RS | | | RD | | | RDS | | |
| Methods | PSNR | SSIM | LPIPS | PSNR | SSIM | LPIPS | PSNR | SSIM | LPIPS | PSNR | SSIM | LPIPS | PSNR | SSIM | LPIPS |
| PReNet | 27.1939 | 0.8881 | 0.3941 | 23.4536 | 0.7977 | 0.5036 | 23.0181 | 0.6857 | 0.3267 | 19.5145 | 0.6270 | 0.3980 | 18.7119 | 0.5900 | 0.4454 |
| RCDNet | 27.1157 | 0.8862 | 0.3965 | 23.4234 | 0.7964 | 0.5050 | 23.6687 | 0.6763 | 0.3540 | 21.5567 | 0.6246 | 0.4106 | 20.6816 | 0.5859 | 0.4615 |
| MPRNet | 27.1221 | 0.8867 | 0.4007 | 23.5270 | 0.7933 | 0.5097 | 23.9263 | 0.6872 | 0.3231 | 21.9558 | 0.6339 | 0.3831 | 21.0407 | 0.5972 | 0.4338 |
| IDT | 26.9428 | 0.8873 | 0.3912 | 23.4492 | 0.7997 | 0.4977 | 24.1806 | **0.7088** | 0.2950 | 21.8945 | **0.6551** | 0.3635 | 21.0991 | **0.6219** | 0.4021 |
| SFNet | 26.7338 | 0.8861 | 0.3912 | 23.2136 | 0.7984 | 0.4964 | 24.4064 | 0.6971 | 0.3000 | 22.0831 | 0.6510 | 0.3521 | 21.0251 | 0.6095 | 0.4159 |
| DRSformer | 27.2100 | 0.8885 | 0.3932 | 23.7299 | **0.8049** | 0.4970 | 24.8096 | 0.7052 | **0.2833** | 21.7949 | 0.6415 | 0.3658 | 20.7358 | 0.6040 | 0.4046 |
| RLP | 26.8646 | 0.8801 | 0.4026 | 23.1733 | 0.7898 | 0.5119 | 22.7828 | 0.6601 | 0.3411 | 20.5325 | 0.6136 | 0.3909 | 19.6163 | 0.5694 | 0.4521 |
| MSGNN | 25.5384 | 0.8692 | 0.4337 | 22.0136 | 0.7702 | 0.5354 | 22.7039 | 0.6572 | 0.3427 | 19.3446 | 0.6168 | 0.3735 | 18.2088 | 0.5637 | 0.4476 |
| NeRD-Rain | 27.1613 | **0.8895** | 0.3867 | 23.6547 | 0.8046 | 0.4915 | 24.2879 | 0.6870 | 0.2912 | 22.0290 | 0.6329 | 0.3523 | 21.1359 | 0.5943 | 0.4095 |
| CST-Net (Ours) | **27.3064** | 0.8891 | **0.3805** | **23.8114** | 0.8062 | **0.4877** | **25.0456** | 0.7065 | 0.2715 | **22.7280** | 0.6499 | **0.3479** | **22.0070** | 0.6175 | **0.3816** |

## 4.3 Implicit Illumination Guidance

To capture illumination information to guide the nighttime rain removal, we introduce an Implicit Illumination Guidance (IIG) module between these two stages. Different from existing approaches that rely on explicit illumination models to achieve illumination estimation [1], we integrate the implicit neural representation into our model to better encode illumination information in complex nighttime scenes.

Specifically, we first leverage a shared encoder from the degradation removal stage to encode features $E$. The pixel coordinates of each nighttime rainy patch $I_{Y'}$ are stored in the corresponding coordinate set $P \in \mathbb{R}^{H \times W \times 2}$, where the value '2' represents horizontal and vertical coordinates. Given the uneven illumination in nighttime images, we apply dynamic weights for each pixel coordinate and its corresponding features, defined as follows:

$$P_{(x,y)} = \sum_{(x',y') \in \Omega(x,y)} w(x',y') I_{Y'}(x',y'), \tag{11}$$

where $P_{(x,y)}$ represents the local illumination information at the central coordinate $(x, y)$, while $\Omega(x,y)$ denotes the neighborhood region centered around $(x, y)$; $w(x', y')$ is a weight for illumination focusing, with the weight decreasing as the distance from the central pixel increases.

By predicting the Y channel value of each pixel, we fuse encode features $E$ and $P_{(x,y)}$, then feed them into the MLP as a decoding function to obtain the final output image $I_{\hat{Y}}$, which is defined as:

$$I_{\hat{Y}} = \text{MLP}(cat[E, P_{(x,y)}]), \tag{12}$$

where the MLP is adopted to map coordinates to their predicted Y values, instead of the RGB values used in existing works [6, 44]. Here, fitting the implicit neural representation to reconstruct an image requires finding a set of parameters for the MLP. Diverse illumination distributions yield different sets of parameters, which in turn means the MLP is adaptive to the complex nighttime rainy scenes.

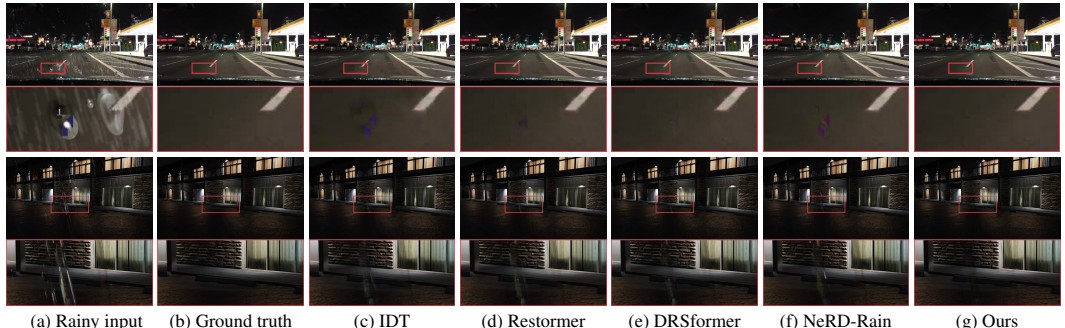

| (a) Rainy input | (b) Ground truth | (c) IDT | (d) Restormer | (e) DRSformer | (f) NeRD-Rain | (g) Ours |

Figure 5: Derained results on the HQ-NightRain (first row) and GTAV-NightRain [53] (second row).

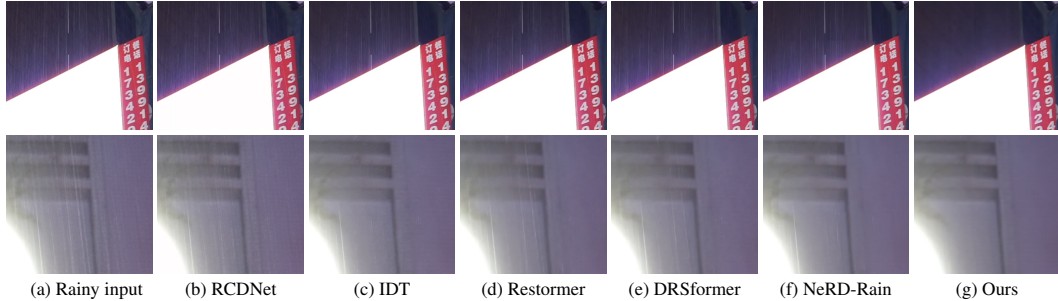

| (a) Rainy input | (b) RCDNet | (c) IDT | (d) Restormer | (e) DRSformer | (f) NeRD-Rain | (g) Ours |

Figure 6: Derained results on the real-world nighttime rainy image from the RealRain-1k [28] dataset.

## 5 Experiments

### 5.1 Experimental Settings

**Datasets and metrics**. We conduct the experiments on our HQ-NightRain dataset, a public nighttime deraining dataset GTAV-NightRain set1 [53], and two real-world benchmarks (*e.g.*, RealRain-1k [28] and RainDS-real [33]). To evaluate the quality of each derained image, we use PSNR [20, 3], SSIM [40], LPIPS [57], PaQ-2-PiQ [49], and MANIQA [45] as evaluation metrics.

**Implementation details**. In our CST-Net, both the degradation removal stage and the color refinement stage adopt a 4-level Transformer-based encoder-decoder structure [51]. We conduct training using an NVIDIA GeForce RTX 3090 GPU. The Adam optimizer [13] with default parameters is used. The initial learning rate is set to $2 \times 10^{-4}$ and gradually reduces to $1 \times 10^{-6}$ using a cosine annealing scheduler. The model is trained for 500 epochs with a patch size of $128 \times 128$ pixels and a batch size of 4. We set the illumination thresholds $\tau_1$ and $\tau_2$ to 0.2 and 0.8, respectively.

### 5.2 Comparison with the state-of-the-art

We compare our method with 11 image deraining technologies, including PReNet [34], RCDNet [37], SPDNet [48], MPRNet [52], IDT [42], Restormer [51], SFNet [12], DRSformer [5], RLP [54], MSGNN [36], and NeRD-Rain [6].

**Evaluations on the HQ-NightRain dataset**. Table 1 reports the quantitative results of different approaches on the HQ-NightRain dataset. All methods are retrained on the proposed dataset for a fair comparison. It can be observed that our proposed CST-Net achieves the highest PSNR and lowest LPIPS values across various types of nighttime rain, demonstrating the effectiveness of our method in nighttime rain removal. Specifically, our method outperforms SOTA method NeRD-Rain [6] by 0.81dB PSNR on the SD subset. In the first row of Figure 5, we further show the visual comparison. Compared to existing methods that still leave residual rain, our method can restore clearer results.

**Evaluations on public datasets**. Table 1 summarizes the quantitative results on the public GTAV-NightRain dataset [53], where our method consistently achieves the best performance. Our method outperforms the NeRD-Rain [6] by 1.1 dB on the GTAV-NightRain dataset [53], demonstrating the

Table 4: Ablation analysis of different variants in our method, including two-stage network pipeline, other color space transformation (RGB, HSV, HSL, YUV and YCbCr), learnable color space converter (CSC), and implicit illumination guidance (IIG).

| Methods | Network Pipeline | | Other Color Space Transformation | | | | | CSC | | IIG | Metrics | |
|---|---|---|---|---|---|---|---|---|---|---|---|---|
| | Stage1 | Stage2 | RGB | HSV | HSL | YUV | YCbCr | Fixed | Learnable | | PSNR | SSIM |
| Ours$_{w/o\ Stage2\&w/\ YCbCr\&w/\ CSC\&w/o\ IIG}$ | ✔ | ✗ | ✗ | ✗ | ✗ | ✗ | ✔ | ✗ | ✔ | ✗ | 35.0858 | 0.9650 |
| Ours$_{w/o\ Stage1\&w/\ YCbCr\&w/\ CSC\&w/o\ IIG}$ | ✗ | ✔ | ✗ | ✗ | ✗ | ✗ | ✔ | ✗ | ✔ | ✗ | 36.4385 | 0.9740 |
| Ours$_{w/\ Stage1+2\&w/\ YCbCr\&w/\ CSC\&w/o\ IIG}$ | ✔ | ✔ | ✗ | ✗ | ✗ | ✗ | ✔ | ✗ | ✔ | ✗ | 39.8767 | 0.9866 |
| Ours$_{w/\ Stage1+2\&w/\ RGB\&w/o\ CSC\&w/o\ IIG}$ | ✔ | ✔ | ✔ | ✗ | ✗ | ✗ | ✗ | ✗ | ✗ | ✗ | 38.7507 | 0.9838 |
| Ours$_{w/\ Stage1+2\&w/\ HSV\&w/o\ CSC\&w/o\ IIG}$ | ✔ | ✔ | ✗ | ✔ | ✗ | ✗ | ✗ | ✔ | ✗ | ✗ | 39.0317 | 0.9843 |
| Ours$_{w/\ Stage1+2\&w/\ HSL\&w/o\ CSC\&w/o\ IIG}$ | ✔ | ✔ | ✗ | ✗ | ✔ | ✗ | ✗ | ✔ | ✗ | ✗ | 39.1613 | 0.9844 |
| Ours$_{w/\ Stage1+2\&w/\ YUV\&w/o\ CSC\&w/o\ IIG}$ | ✔ | ✔ | ✗ | ✗ | ✗ | ✔ | ✗ | ✔ | ✗ | ✗ | 39.1932 | 0.9846 |
| Ours$_{w/\ Stage1+2\&w/\ YCbCr\&w/o\ CSC\&w/o\ IIG}$ | ✔ | ✔ | ✗ | ✗ | ✗ | ✗ | ✔ | ✔ | ✗ | ✗ | 39.5959 | 0.9857 |
| Ours | ✔ | ✔ | ✗ | ✗ | ✗ | ✗ | ✔ | ✗ | ✔ | ✔ | **40.4984** | **0.9881** |

advantage of performing rain removal in the Y channel. As shown in the second row of Figure 5, our method successfully removes rain streaks and restores a clear background.

**Evaluations on real-world datasets**. We also evaluate our method on the challenging RealRain-1k [28] and RainDS-real datasets [33]. Quantitative results in Table 2 demonstrate that our method can effectively handle diverse types of spatially-varying real rain streaks. Figure 6 shows visual comparisons of the evaluated methods, where most deraining methods are sensitive to complex rain streaks in real scenes. Our proposed approach effectively removes nighttime rain, demonstrating better generalization on real-world data.

## 6 Analysis and Discussion

**Real-world generalization with our HQ-NightRain**.

To demonstrate that the model trained on our HQ-NightRain dataset generalizes better to real-world nighttime scenes, we compare the performance of the same model trained on the proposed HQ-NightRain dataset and other synthetic nighttime rainy datasets [35, 53], then tested on real-world rainy images, as shown in Table 3. As presented in Figure 8, the model trained in HQ-NightRain performs better in real-world images, suggesting that our dataset effectively reduces the domain gap between synthetic and real-world nighttime rainy scenes.

Table 3: Performance comparison of other method (*e.g.*, IDT [42]) train on different synthetic nighttime image deraining datasets and test on the real-world dataset RealRain-1k-L [28].

| Training Datasets | PSNR / SSIM |
|---|---|
| GTAV-NightRain | 26.47 / 0.8640 |
| RoadScene-rain | 25.63 / 0.8408 |
| HQ-NightRain (Ours) | **26.94 / 0.8873** |

**Effectiveness of the learnable CSC**.

As shown in Table 4, the CSC ablation study demonstrates its effectiveness. Figure 7 shows that traditional fixed-weight transformations lead to pixel loss in highlight regions during model processing, while our learnable CSC dynamically adjusts channel weight allocation to adapt to complex scenes. Figure 9 shows that our method effectively removes complex real-world rain, while other methods exhibit varying degrees of rain residue.

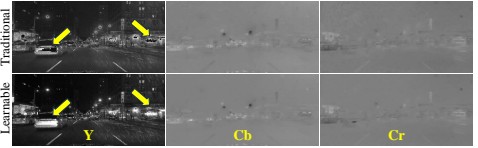

Figure 7: Visualization of intermediate features in the degradation removal stage.

**Evaluations on other color space**. To demonstrate the effectiveness of performing deraining in the YCbCr space, we conduct ablation analysis for different color spaces (*e.g.*, RGB, HSV, HSL, YUV, and YCbCr). As shown in Table 4, it is evident that our model (in YCbCr space) achieves the highest results, as nighttime rain is more prominent in the Y channel, which better facilitates deraining.

**Effectiveness of network components**.

To validate the effectiveness of our network components, we conduct ablation studies in Table 4. All variant models are trained and tested on the HQ-NightRain dataset. The results validate the effectiveness of our components. As shown in the visualization results of Figure 10, the CSC

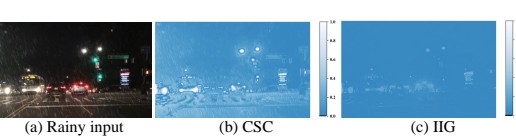

Figure 10: Feature visualization from network components.

Table 5: Performance comparison of nighttime deraining in natural scenes using non-reference metrics.

| Methods | PaQ-2-PiQ / MANIQA |
|---|---|
| IDT | 63.4735 / 0.5056 |
| Restormer | 63.7980 / 0.5010 |
| MSGNN | 56.8076 / 0.4483 |
| CST-Net (Ours) | **63.8756 / 0.5079** |

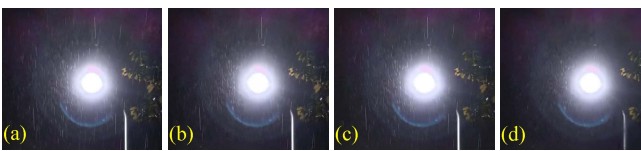

(a)  (b)  (c)  (d)

Figure 8: Generalization results on real-world nighttime rainy images trained on different nighttime image deraining datasets. Real rainy input (a), and results from models trained on GTAV-NightRain (b), RoadScene-rain (c), and HQ-NightRain (d).

Table 6: Performance comparison of multi-weather restoration on the Multi-Weather6K dataset [29].

| Methods | PSNR / SSIM |
|---|---|
| Restormer | 31.80 / 0.9228 |
| TransWeather | 30.75 / 0.9468 |
| PromptIR | 31.69 / 0.9169 |
| CST-Net (Ours) | **33.82 / 0.9642** |

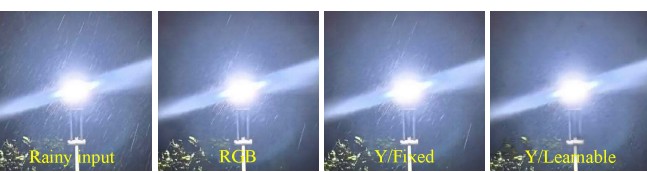

Rainy Input  RGB  Y/Fixed  Y/Learnable

Figure 9: Visual comparisons on real-world nighttime rainy images. 'RGB' denotes processing in the RGB color space, 'Y' denotes processing in the Y channel, 'Fixed' 'Learnable' represent CSC with fixed weights and CSC with learnable weights.

demonstrates excellent rain streak feature extraction capabilities, while the IIG focuses on illuminated regions to guide nighttime deraining.

**Extension to multi-weather restoration**. We extend our method to multi-weather restoration. Table 6 demonstrates that our method achieves competitive performance on the Muti-Weather6k dataset [29], indicating its potential for multi-weather restoration.

**Effectiveness of data synthesis pipeline components on model performance**. To evaluate the impact of each component in the data synthesis pipeline on the final model performance, we conduct an analysis using CST-Net, and the results are shown in Table 7. The limited performance of variant D1 indicates that simple linear addition is insufficient for generating realistic images. Our pipeline, by simulating more challenging nighttime scenarios, achieves a balanced improvement in performance.

Table 7: Impact of dataset synthesis pipeline components on model performance, where Linear denotes linear addition, $f[\cdot]$ denotes convolutional merge, $\sigma[\cdot]$ denotes illumination merge, and $\rho[\cdot]$ denotes defocus blur.

| Variants | Linear | $f[\cdot]$ | $\sigma[\cdot]$ | $\rho[\cdot]$ | PSNR | SSIM |
|---|---|---|---|---|---|---|
| D1 | ✔ | ✘ | ✘ | ✘ | 25.38 | 0.8705 |
| D2 | ✘ | ✔ | ✘ | ✘ | 28.58 | 0.9456 |
| D3 | ✘ | ✔ | ✔ | ✘ | 36.55 | 0.9754 |
| Ours | ✘ | ✔ | ✔ | ✔ | 31.91 | 0.9493 |

**Extension to natural-scene rain removal**. To further validate the effectiveness of our method in nighttime rain removal under natural scenes, we conduct experiments on the Nature subset of the HQ-NightRain dataset. Generalization testing is performed using the model trained on the RS subset, and the quantitative results based on non-reference metrics are shown in Table 5. The proposed method shows consistently better scores across multiple non-reference metrics, indicating enhanced perceptual quality and stronger robustness in complex natural scenes.

**Application**. To evaluate the applicability of our method to outdoor vision tasks such as object detection, we test it on BDD350-Night [21] using a pre-trained YOLOv8 model. Our method achieves the highest precision, recall, and IoU, showing strong potential for downstream tasks. Beyond denoising, we also apply our data synthesis pipeline to film and game production, generating realistic rainy scenes by modeling illumination effects.

## 7 Conclusion

In this paper, we rethink nighttime deraining task and propose HQ-NightRain, a high-quality benchmark that reduces the domain gap between synthetic and real data. We also introduce a color space transformation framework to enhance rain removal in the Y channel. Extensive experiments show that our method outperforms state-of-the-art approaches on both synthetic and real benchmarks.

## Acknowledgments and Disclosure of Funding

This work was supported in part by the Scientific Research Project of the Education Department of Liaoning Province (No. LJ212410152006).

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

# A    Technical Appendices and Supplementary Material

**Overview.** The supplementary includes the following sections:

# B    Motivation of the Method

In this section, we explore the histogram characteristics of different color space channels in daytime and nighttime scenes, as shown in Figures 11 and 12.

In **daytime scenes**, the histogram difference between rainy and rain-free images in the Y channel is relatively small. This is primarily due to the presence of sufficient illumination, which ensures a high and evenly distributed scene brightness. Natural daylight allows a wide range of brightness levels, resulting in pixel values being more uniformly distributed within the 0-255 range. Consequently, rain streaks have a limited impact on the overall brightness distribution. Additionally, rain streaks mainly cause local contrast variations, which, under well-balanced daylight illumination, do not significantly alter the overall histogram distribution.

In contrast, **nighttime scenes** exhibit a more pronounced difference between the histograms of rainy and rain-free images in the Y channel. Due to weaker illumination at night, most pixel values are concentrated within a lower brightness range (0-100), while high-brightness areas are restricted to regions illuminated by artificial light sources such as streetlights and vehicle headlights. Under these low-light conditions, rain streaks substantially alter local illumination characteristics, leading to a significant shift in the pixel value distribution of the Y channel. Moreover, the scattering and blurring effects induced by rain streaks under illumination further amplify brightness variations, resulting in a more noticeable distinction between the histograms of rainy and rain-free images. This observation suggests that, in nighttime image processing, the Y channel effectively captures rain streak features, providing a strong foundation for brightness-aware deraining methods.

Beyond the YCbCr color space, the histogram variations in the HSV and LAB color spaces also reflect the impact of rain streaks on different image channels. In the **HSV color space**, the V channel represents image brightness and exhibits a distribution similar to that of the Y channel. However, due to the nonlinear nature of brightness representation in HSV, its response to rain streaks is less pronounced than that of YCbCr in certain scenarios. In the **LAB color space**, the L channel also encodes brightness information, but as LAB is designed for perceptual uniformity, its brightness distribution does not highlight rain streak-induced differences as prominently as YCbCr. Overall, among the explored color spaces, the Y channel in YCbCr demonstrates the most distinct response to rain streaks in nighttime scenes, making it a particularly effective choice for nighttime deraining tasks, whereas HSV and LAB show relatively weaker advantages in this regard.

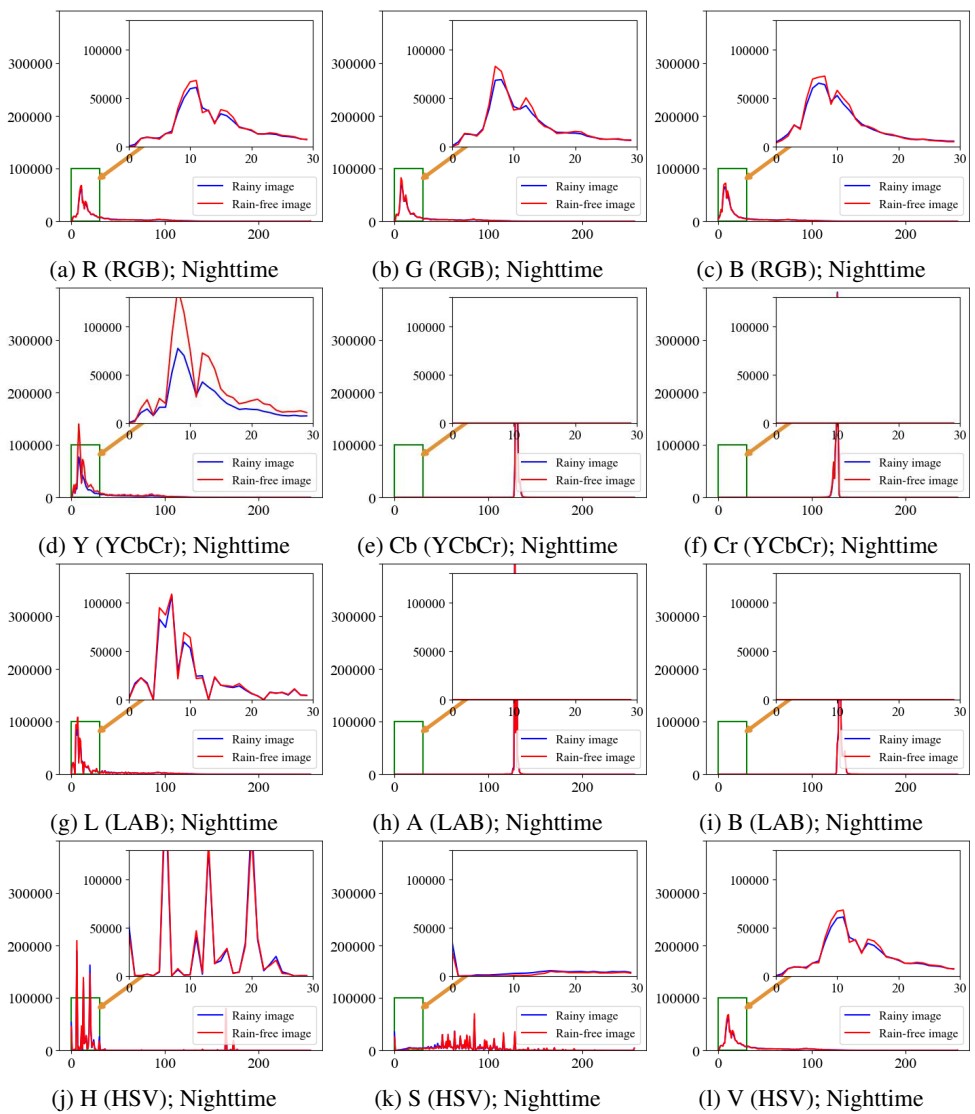

Figure 11: Histograms of channels in different color spaces for **nighttime** scenes.

## C Generation of Initial Rain Streak Masks and Raindrop Masks

**Generation of rain streak masks.** Rain, as a complex atmospheric phenomenon, is influenced by the combined effects of various natural factors, including raindrop size and density. Fidelity and diversity represent two essential aspects in the process of rain synthesis [7]. Inspired by [15, 39, 7], we model the generation of rain streak layers as a motion blur process, inherently capturing two key characteristics of rain streaks: repeatability and directionality. The mathematical formulation is defined as follows:

$$\mathbf{S} = \mathcal{T}(\mathbf{K}(l, \theta, \omega) * \mathbf{N}(n)), \tag{13}$$

where $\mathbf{N}$ denotes the rain mask generated by random noise n. We utilize uniform random numbers and thresholding to control the noise level, with $l$ and $\theta$ representing the length and angle of the motion blur kernel $\mathbf{K} \in \mathbb{R}^{p \times p}$, respectively. Subsequently, we apply Gaussian blur to introduce a rotated diagonal kernel, controlling the rain width $w$. Finally, the transparency of the rain is controlled using the function $\mathcal{T}$, generating the final rain mask $\mathbf{S}$. The noise quantity $n$, rain length $l$, rain angle $\theta$, and rain thickness $w$ are obtained by sampling from $[50, 200]$, $[20, 50]$, $[-30°, 30°]$, and $[3, 7]$.

**Generation of raindrop masks.** Inspired by work [2], in order to achieve higher quality and more realistic raindrop synthesis images, we use the open-source 3D graphics engine (Blender) to simulate

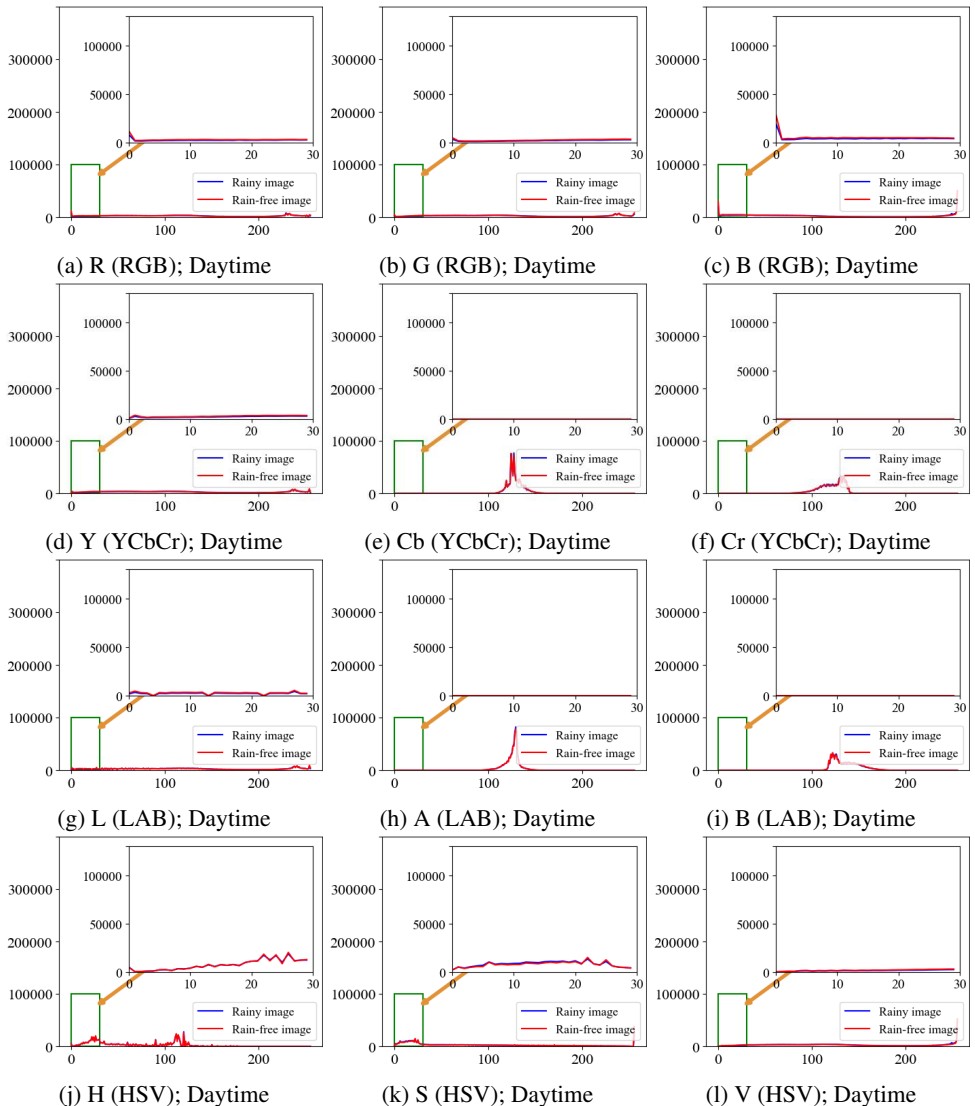

Figure 12: Histograms of channels in different color spaces for **daytime** scenes.

and generate real raindrop images. This 3D graphics engine can render raindrops using a physical motion model, allowing us to set depth information and color values separately in the RGB channels [19], facilitated by a Blender plugin called Rain Generator. Inspired by the work in [2], we model the generation of the raindrop layer as a motion blur process. The instantaneous shape of the raindrop at time $t$ is represented by the function $r[t, \theta, \phi]$, where $r$ is the distance from the surface of the droplet to its center, $\theta$ is the angle defined as the opposite direction to the point where the droplet falls, and $\phi$ is the angle between the point and the projection of the line of sight onto any plane perpendicular to the fall direction. The mathematical definition is as follows:

$$r[t, \theta, \phi] = r_0(1 + \sum_m cos(m\phi)p_m(\theta)), \tag{14}$$

where $r_0$ represents the undeformed radius of the raindrop, and the factor $cos(m\phi)$ depends on the droplet size $r_0$. The function $p_m(\theta)$ describes the time-dependent variation of the modal shape and amplitude relative to $\theta$. As the raindrop falls, the effects of aerodynamic forces and surface tension acting on the droplet lead to rapid shape distortions over time.

# D Pseudo-code for Dataset Synthesis

We represent the dataset synthesis process using pseudocode, as shown in Algorithm 1.

---

**Algorithm 1** The pseudocode of our proposed framework for synthesizing datasets.

---

1: **Input:** Background $B$, $type$ (Rain streak or Raindrop)
2: **Output:** Rain degradation image ($O_{RS}$ or $O_{RD}$)
3: **procedure** GENERATEILLUMINANCEMATRIX($B$)
4:     $B_{HSV} \leftarrow$ RGBToHSV($B$)
5:     $N \leftarrow$ Normalize(ExtractVChannel($B_{HSV}$))
6:     $I \leftarrow \delta(N)$
7:     **return** $I$
8: **end procedure**

9: **procedure** GENERATEMASKS($B$, $type$)
10:     **if** $type$ = 'Rain streak' **then**
11:         $RS_1 \leftarrow$ GenerateRainStreakMask($B$)
12:         $M \leftarrow RS_2 \leftarrow \sigma(RS_1)$
13:     **else if** $type$ = 'Raindrop' **then**
14:         $RD_1 \leftarrow$ GenerateRaindropMask($B$)
15:         $M \leftarrow RD_2 \leftarrow \sigma(RD_1)$
16:     **end if**
17:     **return** $M$
18: **end procedure**

19: **procedure** MAINPROCESS($B$, $M$, $type$)
20:     **if** $type$ = 'Rain streak' **then**
21:         $O_{RS} \leftarrow f[B, M]$
22:         **return** $O_{RS}$
23:     **else if** $type$ = 'Raindrop' **then**
24:         $B_{blur} \leftarrow \rho(B)$
25:         $O_{RD} \leftarrow f[B_{blur}, M]$
26:         **return** $O_{RD}$
27:     **end if**
28: **end procedure**

---

# E Overview of the HQ-NightRain Dataset

Table 8 provides a detailed overview of our proposed HQ-NightRain dataset, including the number of images in each subset.

Table 8: Overview of our proposed HQ-NightRain dataset. The dataset includes rain streaks (RS), raindrops (RD), a mixture of rain streaks and raindrops (SD), real nighttime rain images (Real), and natural-scene nighttime rain images (Nature).

| Subset | Number | | | | |
| --- | --- | --- | --- | --- | --- |
| | RS | RD | SD | Real | Nature |
| Training set | 5,000 (pairs) | 2,500 (pairs) | 2,500 (pairs) | / | / |
| Validation set | 500 (pairs) | 200 (pairs) | 200 (pairs) | / | / |
| Testing set | 100 (pairs) | 100 (pairs) | 100 (pairs) | 512 | 20 (pairs) |

# F Dataset Visualization

In this section, we present additional sample images from the proposed HQ-NightRain dataset in Figure 13. It can be observed that our dataset exhibits greater visual realism and harmony.

# G Comparison with Existing Datasets

In this section, we summarize commonly used datasets for daytime and nighttime scenarios, as detailed in Table 9. From the perspective of rain types, the proposed HQ-NightRain dataset offers more

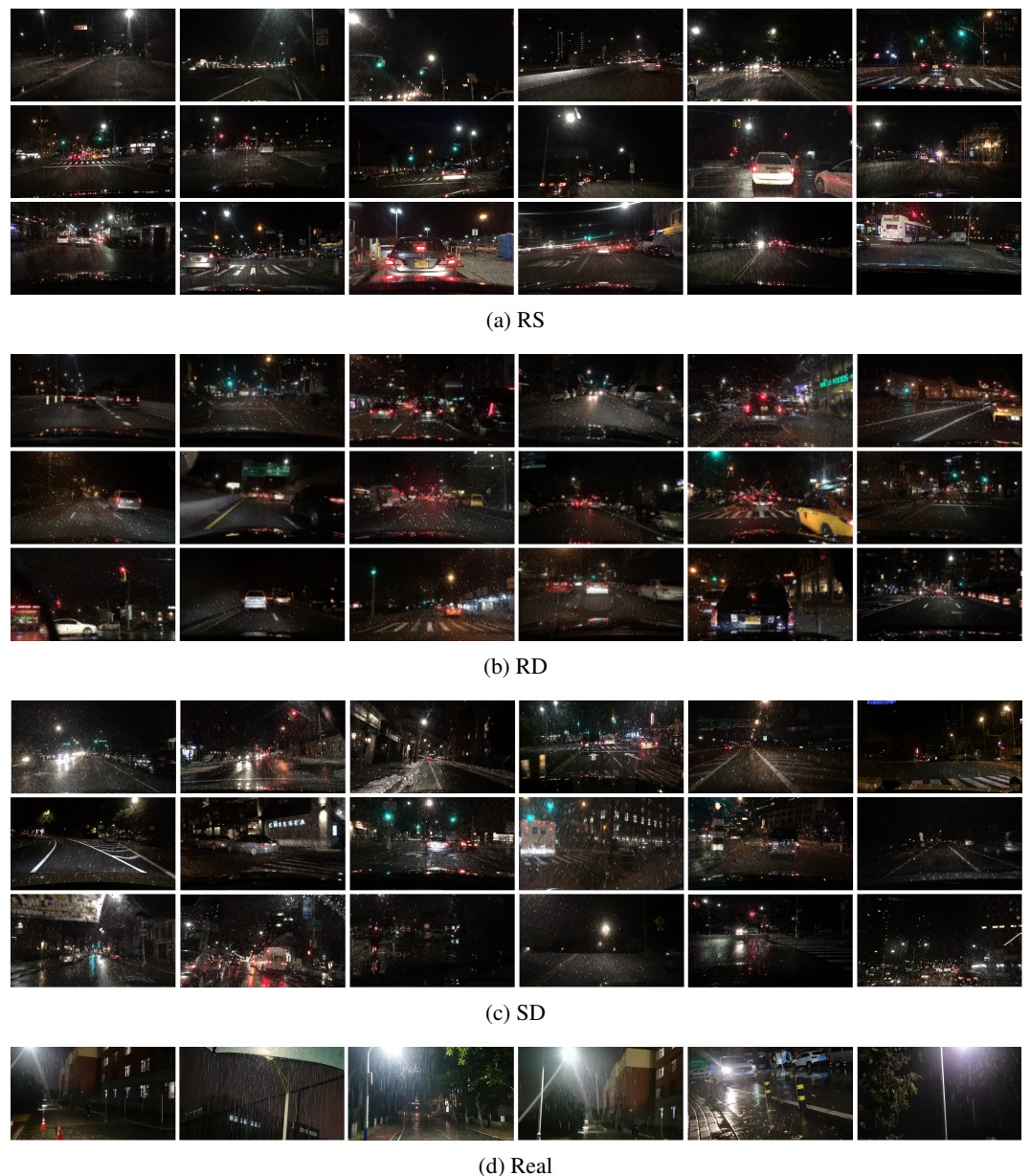

(a) RS

(b) RD

(c) SD

(d) Real

Figure 13: Example images from the HQ-NightRain dataset. The dataset includes rain streaks (RS), raindrops (RD), a mixture of rain streaks and raindrops (SD), and real nighttime rain images (Real).

comprehensive coverage, providing richer data resources for nighttime deraining tasks. Moreover, as our dataset includes corresponding JSON labels, it enables object detection tasks to evaluate the impact of deraining results on downstream applications. We provide additional visual comparisons with other datasets in Figure 14.

Table 9: Comparison of existing daytime and nighttime deraining datasets. Our dataset includes a wider variety of rain types: 'RS' represents rain streaks, 'RD' represents raindrops, 'SD' represents a mixture of raindrops and rain streaks, and 'Real' represents real-world data.

| Type | Datasets | Rain Categories | | | | Annotation |
|------|----------|----|----|----|------|------------|
| | | RS | RD | SD | Real | |
| Daytime | Rain200L/H [46] | ✔ | | | | None |
| | DID/DDN-Data [14, 55] | ✔ | | | | None |
| | Raindrop [32] | | ✔ | | | None |
| | Rain13K [21] | ✔ | | | | None |
| | RainDS [33] | ✔ | ✔ | ✔ | | None |
| | MPID [27] | ✔ | ✔ | | ✔ | Detection |
| Nighttime | GTAV-NightRain [53] | ✔ | | | | None |
| | RoadScene-rain [35] | ✔ | | | | None |
| | Raindrop Clarity [22] | | ✔ | | | None |
| | HQ-NightRain (Ours) | ✔ | ✔ | ✔ | ✔ | Detection |

(a) GTAV-NightRain [53]  (b) RoadScene-rain [35]  (c) HQ-NightRain-RS (Ours)  (d) Raindrop Clarity [22]  (e) HQ-NightRain-RD (Ours)

Figure 14: Further visual comparisons with other nighttime deraining datasets.

## H   User Study

In this section, we conduct two user studies. Our survey process is conducted anonymously, with the images in each set randomly shuffled to ensure fairness. The questionnaire is distributed without restrictions to a broad range of online users, and responses are collected from a total of 72 human evaluators. The first focuses on the illumination thresholds $\tau_1$ and $\tau_2$ used to calculate the illumination coefficient matrix $\mathbf{I}$ in Equation 4 of the main manuscript. Multiple sets of nighttime rain images are generated using various parameter combinations, and users select the images they perceive as most realistic. As shown in Figure 15, images generated with $(\tau_1, \tau_2) = (0.2, 0.8)$ are widely preferred, aligning more closely with realistic visual perception. The second user study focuses on subjective evaluations of the realism of different datasets. We randomly selected several groups of nighttime scene images from various datasets [21, 53, 35, 22], and users were asked to choose the most realistic ones. As shown in Figure 16, the HQ-NightRain dataset was deemed more realistic by the majority of human evaluators.

## I   Details of the Two-Stage Network

Our CST-Net comprises two stages: the degradation removal stage and the color refinement stage. As shown in Figure 17, both stages utilize an identical Transformer-based four-layer encoder-decoder architecture [51]. The expansion ratio of feature channels is set to 2, and the number of modules in the first and second stages is set to $\{N_1', N_2', N_3', N_4'\}$ and $\{N_1'', N_2'', N_3'', N_4''\}$, respectively. Skip-

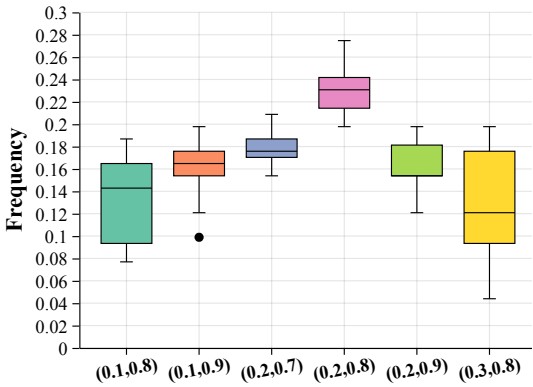

Figure 15: The box plot presents the results of the user study for perceived visual realism scores across different hyperparameters $(\tau_1, \tau_2)$. Higher scores indicate better perceived realism.

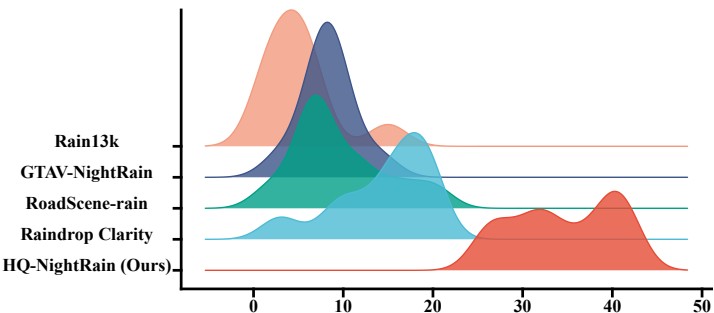

Figure 16: The ridge plot presents the results of the user study, showing the number of perceived realism selections across different datasets.

connections are incorporated to bridge consecutive intermediate features, enabling stable training. The architectures of the modules used in the network are illustrated in Figure 18. Ablation experiments were conducted to evaluate the impact of different module combinations and the number of modules on the network's performance, with results presented in Table 10.

Table 10: Ablation study for different variants of our method includes the normalization techniques, the number of modules at each stage, and the combination method at each stage.

| Methods | LN | BN | $\{N_1', N_2', N_3', N_4'\}$ | $\{N_1'', N_2'', N_3'', N_4''\}$ | Stage1 | Stage2 | PSNR↑ | SSIM↑ | LPIPS↓ |
|---------|----|----|------------------------------|-----------------------------------|--------|--------|-------|-------|--------|
| (a) Ours | ✔ | ✗ | {1,2,2,4} | {1,2,2,4} | SERB+MSFN | MDTA+MSCM | **40.4984** | **0.9881** | **0.0248** |
| (b) | ✔ | ✗ | {1,2,2,4} | {1,2,2,4} | SERB+MSCM | MDTA+MSFN | 39.0333 | 0.9846 | 0.0334 |
| (c) | ✗ | ✔ | {1,2,2,4} | {1,2,2,4} | SERB+MSFN | MDTA+MSCM | 38.8534 | 0.9842 | 0.0336 |
| (d) | ✔ | ✗ | {2,4,4,6} | {2,4,4,6} | SERB+MSFN | MDTA+MSCM | 39.7594 | 0.9861 | 0.0301 |
| (e) | ✔ | ✗ | {4,6,6,8} | {4,6,6,8} | SERB+MSFN | MDTA+MSCM | 40.1056 | 0.9873 | 0.0275 |

## J  Loss Function

In this document, we provide a supplementary introduction to the loss function used. For the degradation removal stage, we utilize the Mean Squared Error (MSE) loss function, defined as follows:

$$\mathcal{L}_{mse} = \frac{1}{n} \sum_{i=1}^{n} \left(Y_{gt} - \bar{Y}\right)^2, \tag{15}$$

where $Y_{gt}$ represents the Y channel of the ground-truth image, and $\bar{Y}$ denotes the predicted result from the degradation removal stage. Additionally, we incorporate Charbonnier [52], Structural Similarity

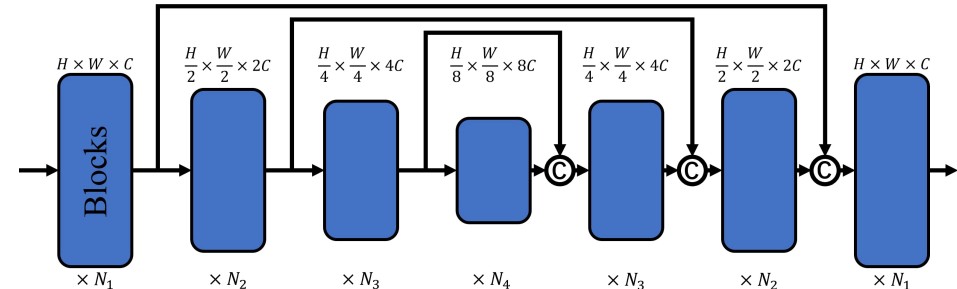

Figure 17: Network architectures of the degradation removal stage and the color refinement stage.

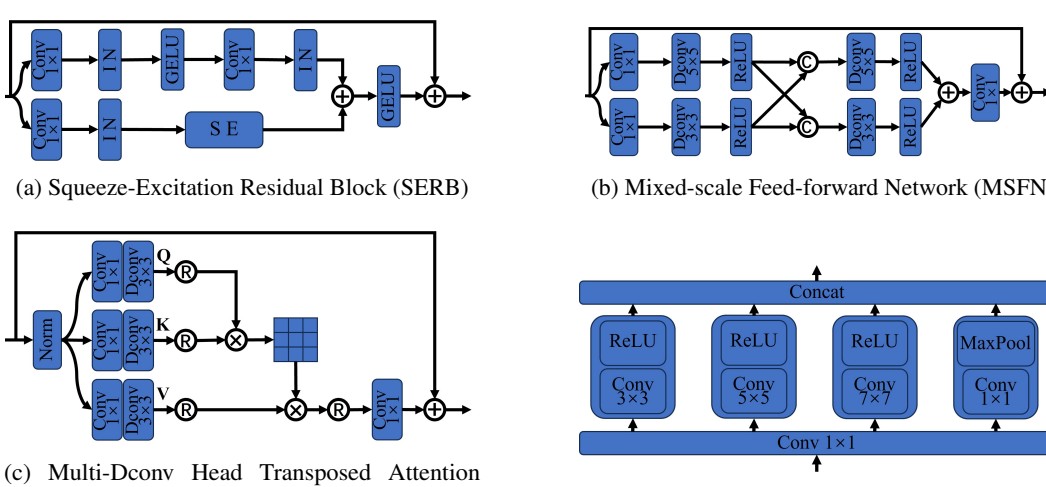

(a) Squeeze-Excitation Residual Block (SERB)

(b) Mixed-scale Feed-forward Network (MSFN)

(c) Multi-Dconv Head Transposed Attention (MDTA)

(d) Multi-Scale Convolutional Module (MSCM)

Figure 18: Detailed structure of the modules used in the network

(SSIM) loss and edge [10] loss to constrain network training. The Charbonnier loss is defined as follows:

$$\mathcal{L}_{char} = \sqrt{\|O_{RGB} - I_{gt}\|^2 + \epsilon^2}, \tag{16}$$

where $O_{RGB}$ denotes the reconstructed image output by the network, $I_{gt}$ represents the ground-truth image and $\epsilon = 10^{-3}$ is an offset value. The structural similarity loss is defined as follows:

$$\mathcal{L}_{ssim} = 1 - \text{SSIM}(O_{RGB} - I_{gt}). \tag{17}$$

The edge loss is defined as follows:

$$\mathcal{L}_{edge} = \frac{\sum_{x=1}^{W} \sum_{y=1}^{H} E_{i,j} \cdot \left( \left| I_{gt_{(i,j)}} - O_{RGB_{(i,j)}} \right| \right)}{WH}, \tag{18}$$

where $W$ and $H$ represent the width and height of the image, respectively. The proposed loss function $\mathcal{L}_{total}$ for network training is defined as follows:

$$\mathcal{L}_{total} = \mathcal{L}_{mse} + \mathcal{L}_{ssim} + \mathcal{L}_{char} + \alpha \cdot \mathcal{L}_{edge}, \tag{19}$$

where $\alpha$ is empirically set to 0.5.

## K   Model Complexity

Table 11 presents the complexity comparison. With a reduced number of modules, our method avoids significant increases in model complexity. Compared to other state-of-the-art methods, our model demonstrates certain advantages.

Table 11: Comparisons of model complexity against state-of-the-art methods. The size of the test image is $256 \times 256$ pixels.

| Methods | RCDNet [37] | MPRNet [52] | Restormer [51] |
|---|---|---|---|
| #FLOPs (G) | 194.502 | 548.652 | 140.990 |
| #Params (M) | 2.958 | 3.637 | 26.097 |
| Methods | NeRD-Rain [6] | DRSformer [5] | CST-Net (Ours) |
| #FLOPs (G) | 147.978 | 220.378 | 144.819 |
| #Params (M) | 22.856 | 33.627 | 16.207 |

## L  Hyperparameter Validation

As shown in Table 12, we validate the illumination threshold hyperparameters $\tau_1$ and $\tau_2$ in Equation 4 of the main manuscript. We empirically test three sets of fixed values and one set of random values. Specifically, we generate datasets using these threshold sets, train them on our CST-Net, and validate them on the RS subset of the HQ-NightRain dataset. The results show that our chosen thresholds (0.2, 0.8) achieve the best performance. Additionally, to evaluate the impact of the thresholds on the dataset's generalization ability, we conduct generalization tests on the real-world dataset RealRain1k-L [28]. The results demonstrate that our settings enhance the realism of the dataset and achieve optimal generalization performance.

Table 12: Illumination threshold hyperparameter verification (PSNR / SSIM).

| $(\tau_1, \tau_2)$ | Random | (0.1, 0.7) | (0.3, 0.9) | (0.2, 0.8) (Ours) |
|---|---|---|---|---|
| HQ-NightRain-RS (Ours) | 40.78 / 0.9909 | 41.91 / 0.9790 | 42.13 / 0.9861 | **42.89 / 0.9924** |
| RealRain1k-L | 26.08 / 0.8758 | 26.56 / 0.8795 | 26.73 / 0.8840 | **27.31 / 0.8891** |

## M  Application of Rain Synthesis Technology in Game Production

Besides the application in image deraining, we also show our application in rain synthesis. Our rain synthesis method effectively incorporates the role of illumination, resulting in a more visually realistic effect. Here, we apply this technique to film and game production to simulate rainfall effects. Figure 19 presents one visual example. Compared to expensive rendering engines used during development [53], our synthesis technique also helps reduce production costs.

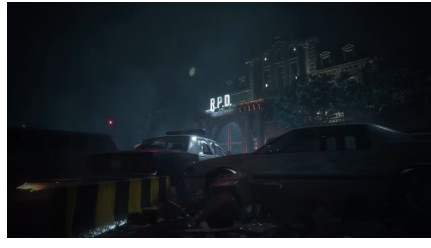
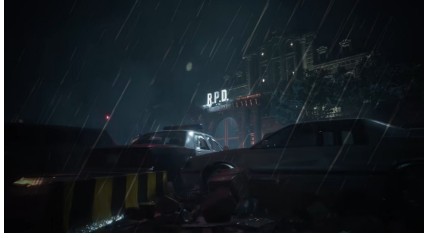

(a) Game screenshot                    (b) Rendering of rain scene

Figure 19: An example of our rain synthesis technique applied to *Resident Evil 2* game scenes.

## N  Limitations

Although our method achieves favorable performance, it fails to handle the veiling effect, especially under low-light conditions where this effect interacts with artificial light sources. Future work will explore the incorporation of physical models to address this issue.

## O  Impact on Downstream Vision Tasks

To investigate the impact of nighttime image deraining on downstream visual tasks, such as object detection, we evaluate on the BDD350-Night [21] dataset using YOLOv8. As presented in Table 13, our results achieve the highest values in Precision, Recall, and IoU across three metrics. As shown in Figures 20 and 21, our deraining results yield higher recognition accuracy, demonstrating that CST-Net effectively enhances subsequent detection performance. Thus, our method has greater potential for application in downstream vision tasks.

Table 13: Performance comparison of joint image deraining, and object detection on the BDD350-Night dataset [21].

| Methods | Rain Input | PReNet | RCDNet | IDT | Restormer | SFNet | DRSformer | RLP | NeRD-Rain | CST-Net (Ours) |
|---|---|---|---|---|---|---|---|---|---|---|
| | Deraining; | | Dataset: **BDD350-Night**; | | Image Size: $1280 \times 720$ | | | | | |
| PSNR↑ | 10.7687 | 11.6005 | 11.7083 | 12.1124 | 12.2404 | 12.0769 | 12.2472 | 11.7422 | 12.1508 | **12.3884** |
| SSIM↑ | 0.1773 | 0.1901 | 0.1967 | 0.2101 | 0.2141 | 0.2229 | 0.2224 | 0.1931 | 0.2101 | **0.2244** |
| | Object Detection; | | Algorithm: **YOLOv8**; | | Dataset: **BDD350-Night**; | | Threshold: 0.6 | | | |
| Precision(%)↑ | 16.00 | 14.72 | 18.71 | 16.20 | 20.24 | 19.43 | 18.37 | 13.69 | 20.19 | **20.49** |
| Recall(%)↑ | 4.66 | 4.66 | 6.21 | 5.63 | 6.60 | 6.60 | 6.99 | 4.47 | **8.16** | **8.16** |
| IoU(%)↑ | 20.26 | 20.34 | 22.27 | 21.36 | 23.08 | 22.77 | 21.56 | 18.42 | 22.27 | **23.29** |

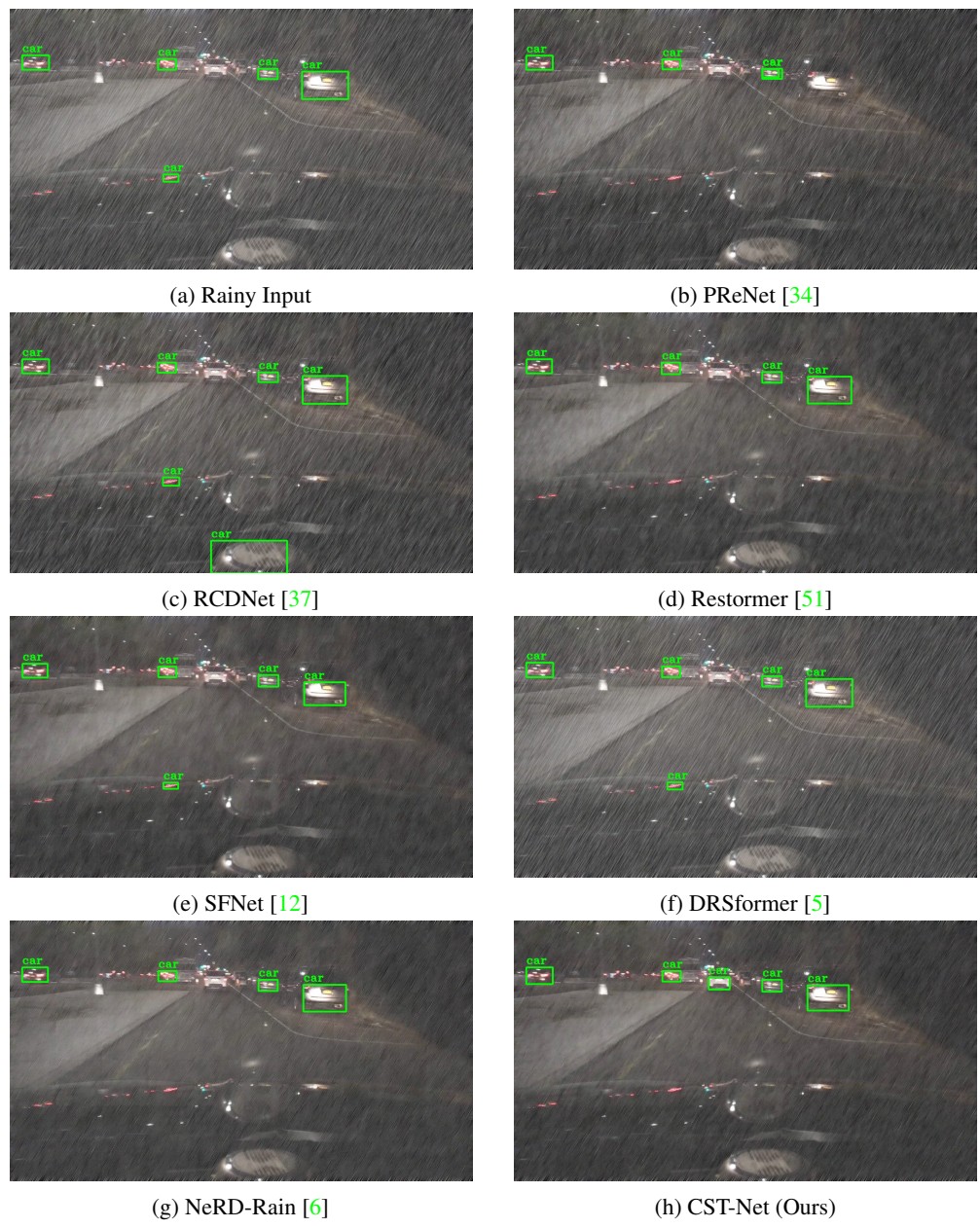

(a) Rainy Input

(b) PReNet [34]

(c) RCDNet [37]

(d) Restormer [51]

(e) SFNet [12]

(f) DRSformer [5]

(g) NeRD-Rain [6]

(h) CST-Net (Ours)

Figure 20: Comparison of image deraining and object detection on the BDD350-Night dataset [21].

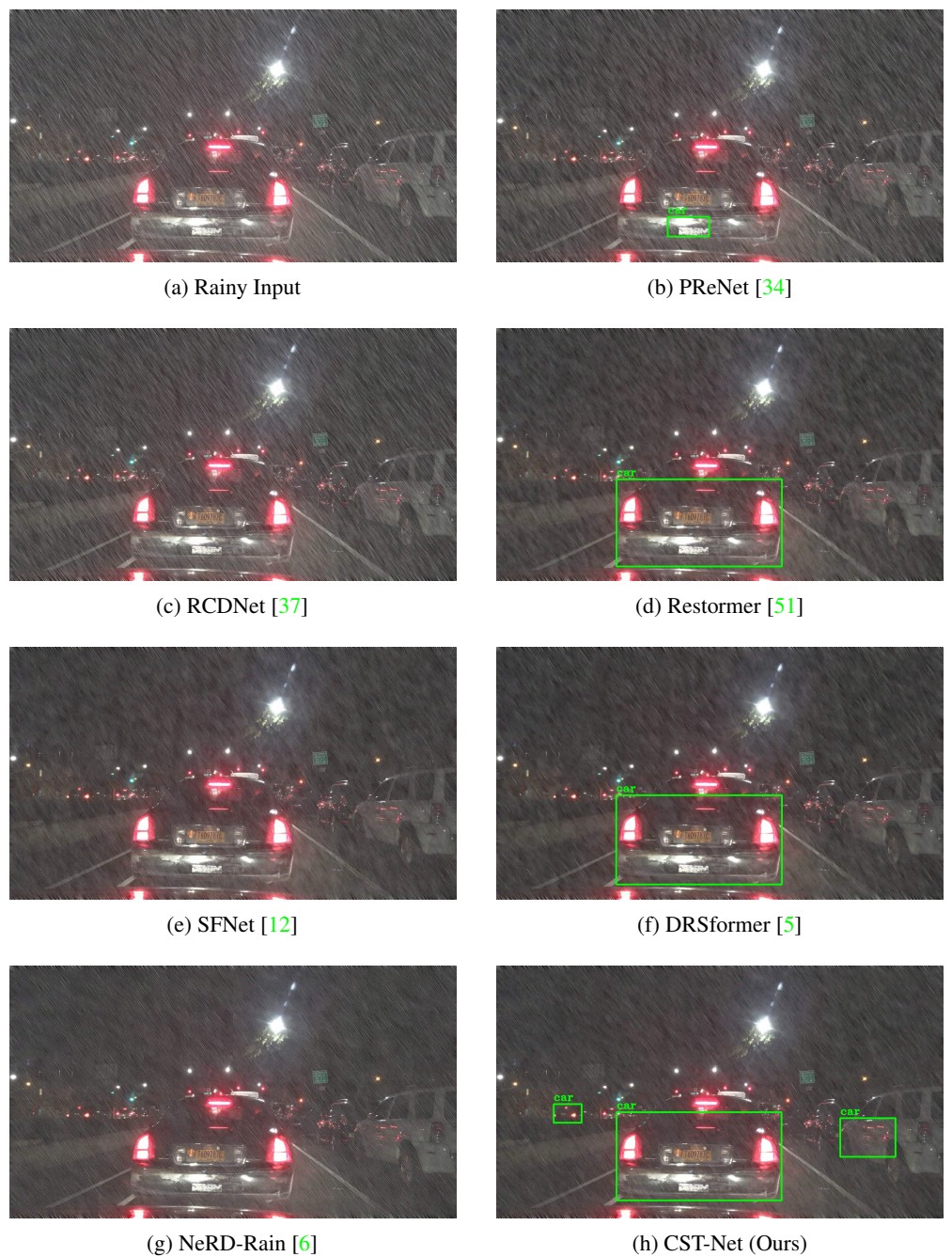

(a) Rainy Input      (b) PReNet [34]

(c) RCDNet [37]      (d) Restormer [51]

(e) SFNet [12]      (f) DRSformer [5]

(g) NeRD-Rain [6]      (h) CST-Net (Ours)

Figure 21: Comparison of image deraining and object detection on the BDD350-Night dataset [21].

# P  More Experimental Results

In this section, we present additional visual comparison results. Figures 22 and 23 show the visual comparisons on the synthetic dataset, GTAV-NightRain [53]. Compared with other methods, our CST-Net generates high-quality deraining results with more accurate detail and color restoration. Figures 24–26 illustrate the visual results on the real-world RainDS-real [33] dataset, including its three subsets: rain streaks (RS), raindrops (RD), and rain streaks mixed with raindrops (RSD). Our method effectively removes complex and random rain streaks and raindrops, achieving visually satisfactory restoration.

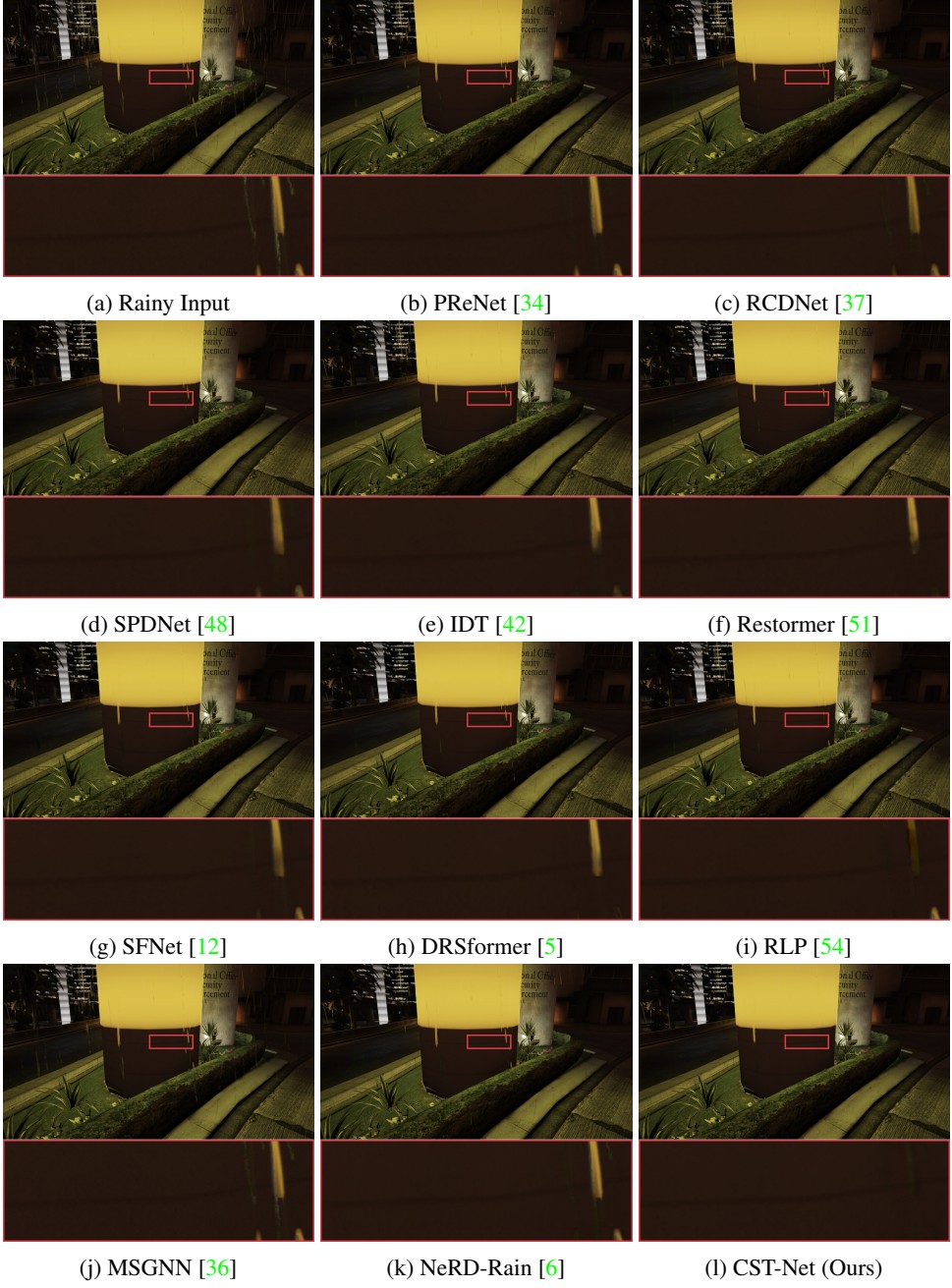

Figure 22: Visual comparison results on the GTAV-NightRain dataset [53]. The results shown in (b)-(k) still contain significant rain streaks. In contrast, our models generate much clearer images.

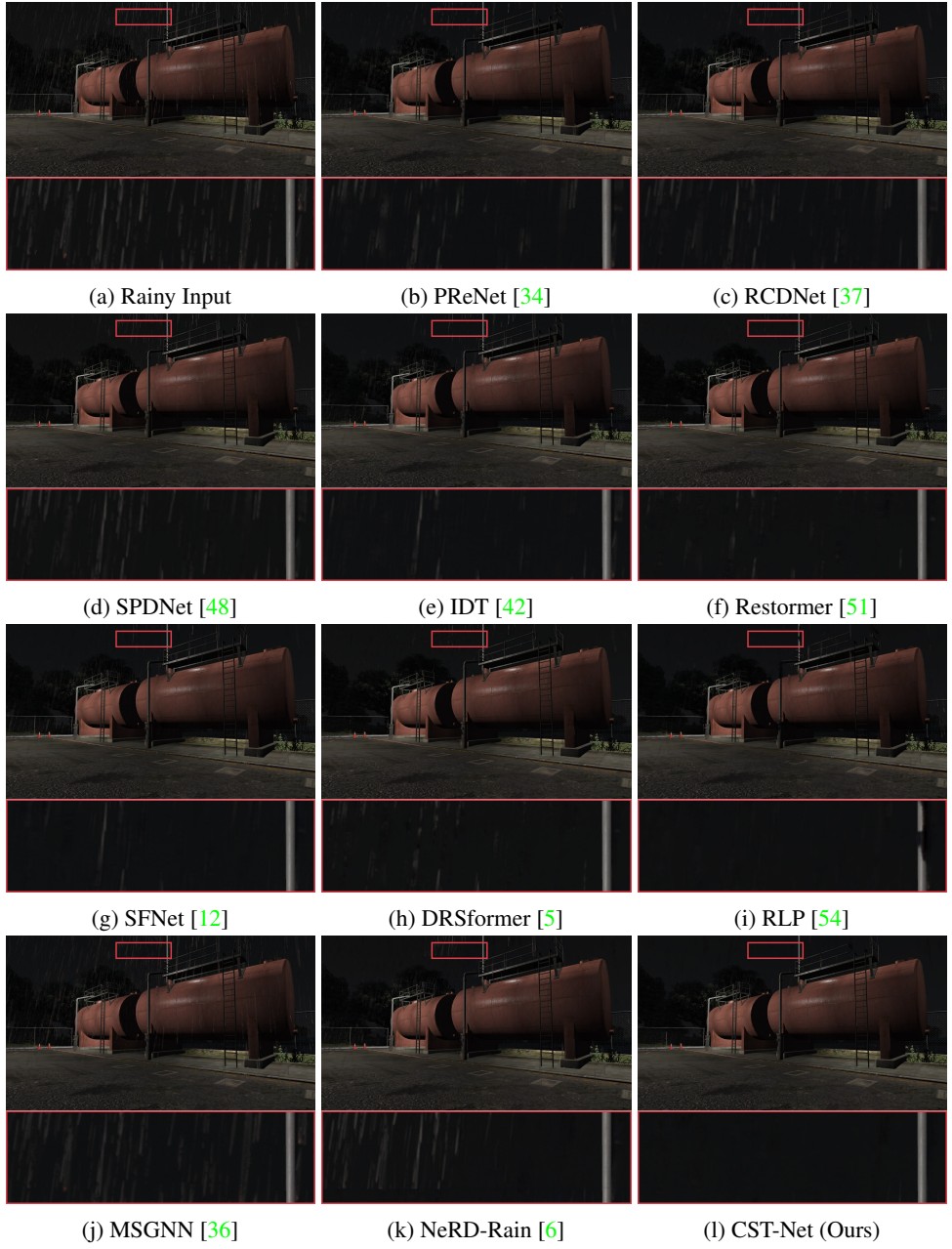

Figure 23: Visual comparison results on the GTAV-NightRain dataset [53]. The results shown in (b)-(k) still contain significant rain streaks. In contrast, our models generate much clearer images.

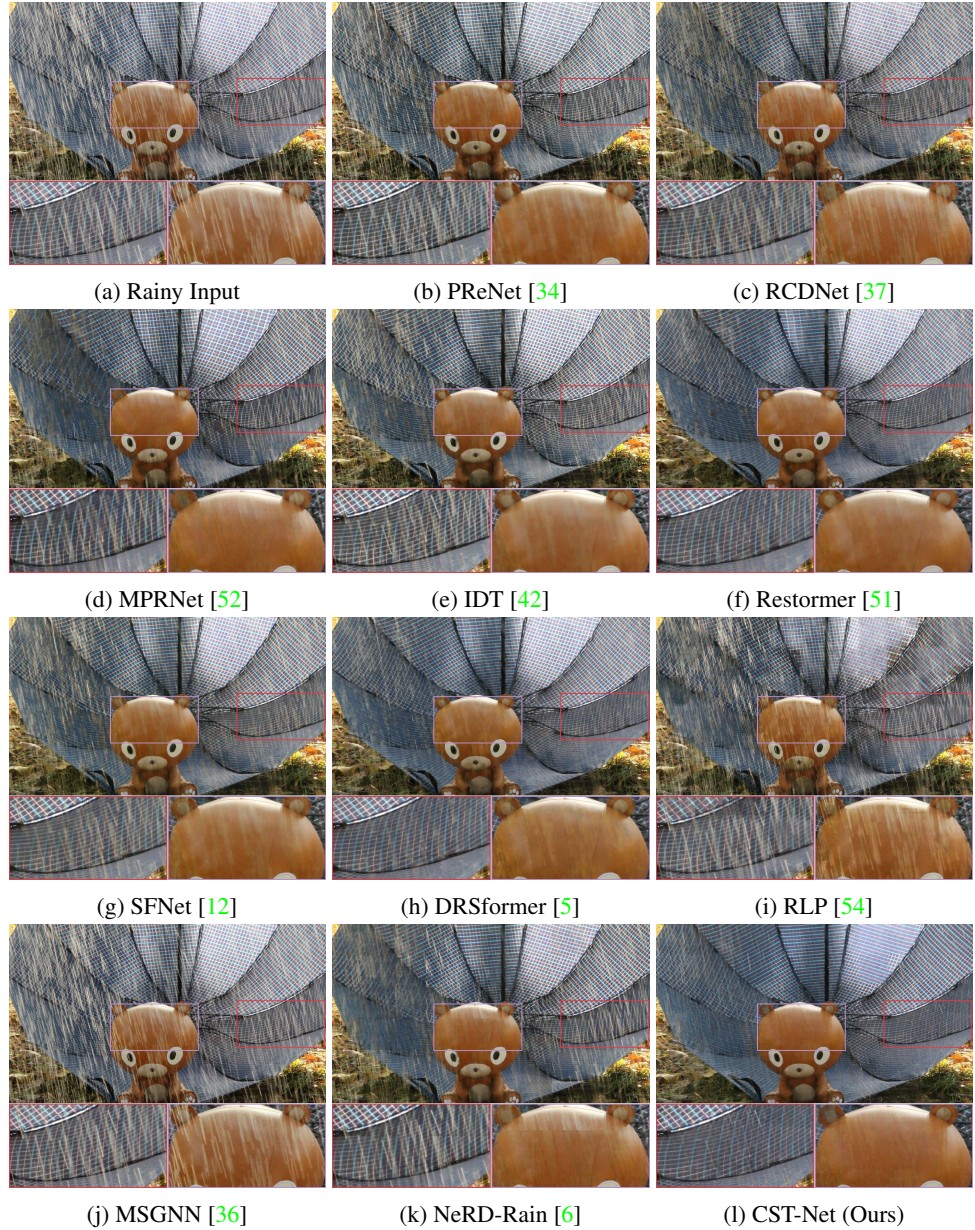

Figure 24: Visual comparison results on the **RS** subset of the RainDS-real dataset [33] reveal that the evaluated methods fail to produce clear images, with some structural details not well restored. In contrast, our method generates derained images with finer structural details and improved clarity.

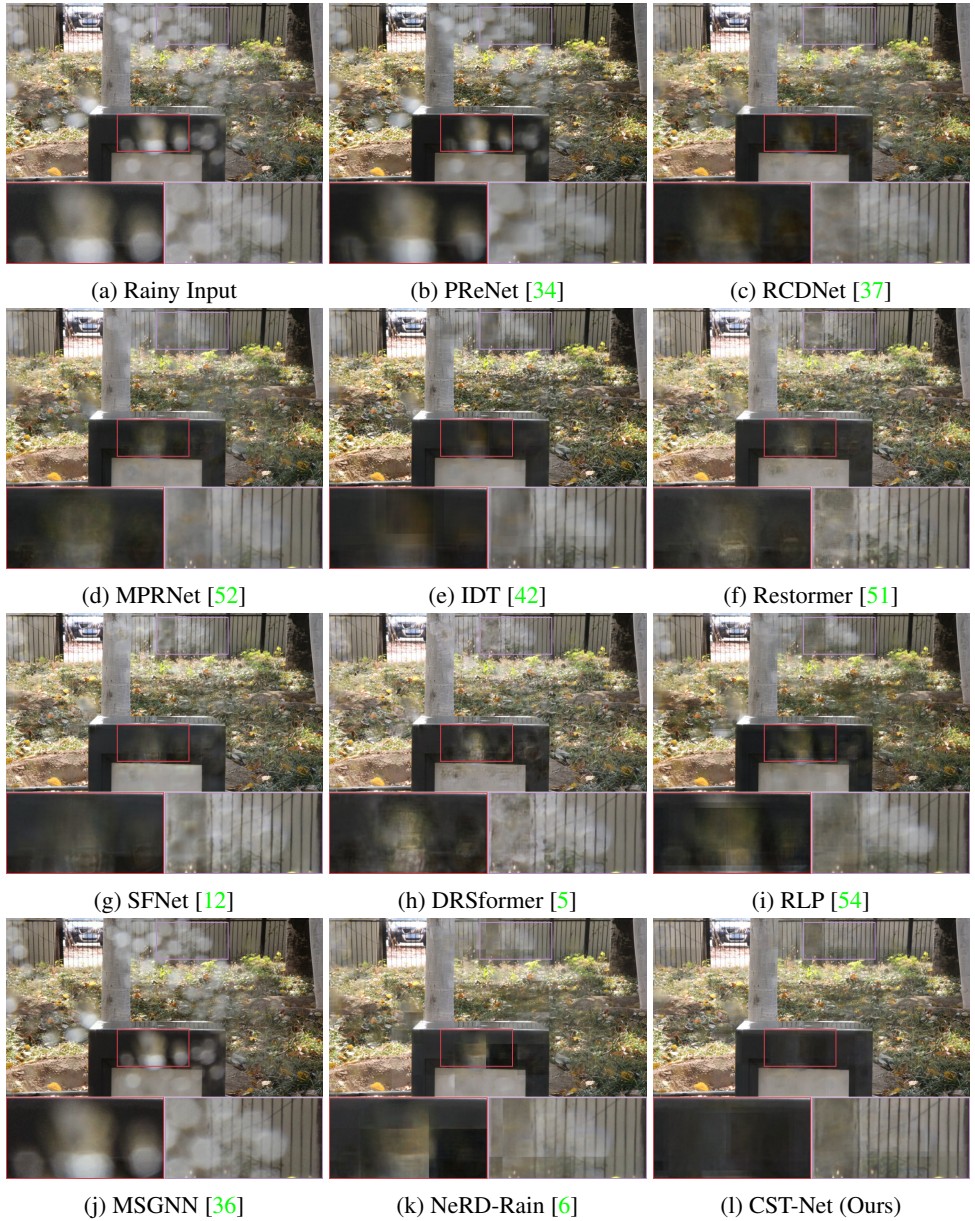

Figure 25: Visual comparison results on the **RD** subset of the RainDS-real dataset [33] reveal that the evaluated methods fail to produce clear images, with some structural details not well restored. In contrast, our method generates derained images with finer structural details and improved clarity.

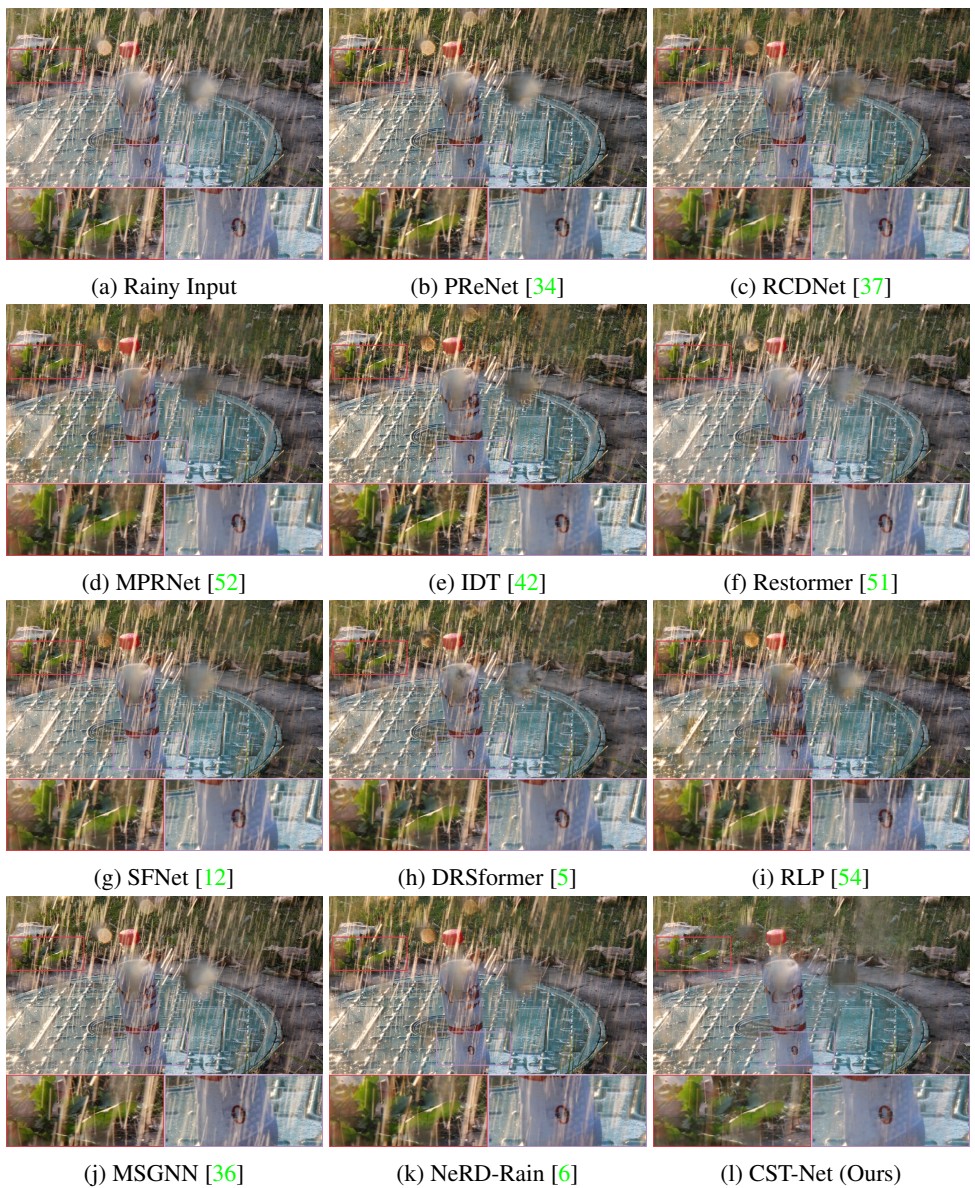

(a) Rainy Input      (b) PReNet [34]      (c) RCDNet [37]

(d) MPRNet [52]      (e) IDT [42]      (f) Restormer [51]

(g) SFNet [12]      (h) DRSformer [5]      (i) RLP [54]

(j) MSGNN [36]      (k) NeRD-Rain [6]      (l) CST-Net (Ours)

Figure 26: Visual comparison results on the **RSD** subset of the RainDS-real dataset [33] reveal that the evaluated methods fail to produce clear images, with some structural details not well restored. In contrast, our method generates derained images with finer structural details and improved clarity.

