# OpenReview forum: "Rethinking Nighttime Image Deraining via Learnable Color Space Transformation"
_NeurIPS.cc/2025/Conference — NeurIPS 2025 poster_

### Official Review · Reviewer_ohMd · 2025-06-21

**Clarity:** 4
**Significance:** 3
**Originality:** 3
**Rating:** 5
**Confidence:** 4

**Summary:**

This paper focuses on the task of nighttime image deraining and introduces a high-quality benchmark dataset HQ-NightRain, which takes into account the impact of night illumination on rain imaging, improves the realism and reduces the domain gap compared with existing datasets. In addition, the article also proposes a learnable color space transformation network (CST-Net). Based on the observation that night rain is more obvious in the Y channel of the YCbCr color space, the YCbCr color space is used for deraining. Extensive evaluations on both synthetic and real datasets demonstrate its excellent performance and generalization ability.

**Questions:**

1. How is the rain mask, which includes both rain streaks and raindrop patterns, generated?
2. It would be helpful if the defocus blur operation used could be explained in more detail. Does this operation rely on the nighttime lighting distribution of the entire image?

**Ethical Concerns:**

["NO or VERY MINOR ethics concerns only"]

**Final Justification:**

After further discussion with the author, I still maintain my acceptance decision.

**Limitations:**

Yes, the authors have discussed the related limitations.

**Quality:**

3

**Strengths And Weaknesses:**

Strengths：
1.  The paper is logically clear and easy to understand, effectively conveying the research.
2.  The HQ-NightRain dataset solves a key problem of nighttime rain removal by considering the effect of illumination on rain imaging, narrowing the domain gap with real data and contributing to the image rain removal community.
3.  Extensive experiments demonstrate that CST-Net outperforms existing solutions on multiple datasets, showing its superior performance in nighttime rain removal. It also shows strong potential in downstream tasks, such as object detection under nighttime rainy conditions, extending its use to more than just rain removal.

Weaknesses：
1. Although the architecture of CST-Net is outlined in the paper, it lacks some implementation details.
2. The proposed method verifies its effectiveness in night scenes. Is it also beneficial in daytime scenes? Can it also effectively remove rain?
3. In the experiments, the Mamba-based image deraining method is missing from the baseline comparisons. It is recommended that the authors include the Mamba-based method to improve the completeness of the benchmark evaluation.

---

> ### Author Rebuttal · Authors · 2025-07-31
>
> We thank Reviewer ohMd for his\her efforts. We are encouraged by the reviewer’s positive comments on the paper presentation, dataset contribution, and superior performance. We answer the questions below and will incorporate all feedback in the revised version.
>
> **1. Details on architecture**
>
> We have provided the implementation details of CST-Net in **Section I of the supplementary materials**.
>
> **2. Effectiveness in daytime scenes**
>
> We would like to claim that we have evaluated our method on the RealRain1k and RainDS-real benchmark datasets, both of which contain rainy images captured in daytime scenes. As shown in the quantitative results in **Table 2 of the main paper**, our method still achieves good performance on both datasets. We have provided visual examples in **Figures 24, 25, and 26 in Section O of the supplementary material**, demonstrating that our model also achieves robust deraining performance in daytime rainy scenes.
>
> **3. Comparison with Mamba-based methods**
>
> According to the reviewer’s suggestion, we further compare our method with recent Mamba-based method, FreqMamba [1]. As shown in the table below, our method outperforms FreqMamba across all three subsets, including rain streaks (RS), raindrops (RD), and the mixture of rain streaks and raindrops (SD). We will add these results in the revised paper.
>
> | Datasets  | HQ-NightRain-RS           | HQ-NightRain-RD           | HQ-NightRain-SD           |
> | :-------- | :------------------------ | :------------------------ | :------------------------ |
> | Methods   | PSNR / SSIM / LPIPS       | PSNR / SSIM / LPIPS       | PSNR / SSIM / LPIPS       |
> | FreqMamba | 42.1469 / 0.9909 / 0.0152 | 33.0119 / 0.9455 / 0.1520 | 37.5566 / 0.9791 / 0.0498 |
> | Ours      | 42.8850 / 0.9924 / 0.0100 | 33.9395 / 0.9523 / 0.1239 | 40.4984 / 0.9881 / 0.0248 |
>
> Ref:
>
> [1] Zhen, Zou, et al. “FreqMamba: Viewing Mamba from a Frequency Perspective for Image Deraining.” ACM MM 2024.
>
>
> **4. Generation of rain streak and raindrop masks**
>
> We have provided detailed descriptions of the generation processes for rain streak masks and raindrop masks in **Section C of the supplementary material**.
>
>
> **5. Issues of the defocus blur operation**
>
> As stated in **L135–138 of the main paper**, we introduce defocus blur based on real-world observations to simulate more realistic nighttime rainy images. The defocus blur is applied to the rain-free nighttime background, without considering the overall nighttime illumination distribution of the image. Illumination distribution is only considered during the synthesis of the rain mask to generate more realistic, non-uniform masks.

---

> > ### Comment · Reviewer_ohMd · 2025-08-03
> >
> > I appreciate the authors’ thoughtful reply, which has resolved my concerns. Having considered the other reviewers’ feedback, I continue to support acceptance of this paper.

---

> > > ### Author Response · Authors · 2025-08-05
> > >
> > > Thank you for your acknowledgment of our work and responses. We appreciate your constructive feedback that has helped refine our research. Please feel free to reach out if you have further queries or need additional clarification on our work.

---

### Official Review · Reviewer_bmcM · 2025-06-21

**Clarity:** 4
**Significance:** 2
**Originality:** 3
**Rating:** 4
**Confidence:** 4

**Summary:**

This paper studies the problem of removing rain from nighttime images. Compared with removing rain during the day, the research on nighttime scenes is more challenging. The contributions of this paper include two aspects. First, it constructs a new dataset HQ-NightRain. On the other hand, it proposes a new image quality restoration framework, color space transformation framework (CST-Net). The method proposed in the paper utilizes the transformation of color space.

**Questions:**

As for this study, I think it is very interesting overall. The paper not only creates a dataset, which is very helpful for nighttime rain removal research. In addition, the paper also proposes a corresponding network based on color space transformation. Some of my doubts are as follows:
(1) Image restoration technology based on color space transformation has been explored in many papers. For example: Guided Real Image Dehazing using YCbCr Color Space.
(2) The proposed network structure and loss function are relatively common according to the supplementary materials. I don't know if I understand it wrongly. The network includes: Squeeze-Excitation Residual Block (SERB), Mixed-scale Feed-forward Network (MSFN, Multi-Dconv Head Transposed Attention (MDTA) and (d) Multi-Scale Convolutional Module (MSCM). This is the main factor for my score.
(3) t-SNE distribution visualization shows the distribution relationship of different datasets. Does this figure show that the proposed nighttime dataset is closer to the real world? How to prove this? I don’t see any overlap in the data points of the proposed dataset and the real world dataset. There is no overlap between the two distribution clusters. In t-SNE, does the distance between different distributions represent the size of their distribution differences?

**Ethical Concerns:**

["NO or VERY MINOR ethics concerns only"]

**Final Justification:**

Based on the author's response and the comments of other reviewers, I gave the final score.

**Limitations:**

yes

**Quality:**

3

**Strengths And Weaknesses:**

Strengths：
(1) This paper constructs a new dataset HQ-NightRain, which is very helpful for night rain removal tasks.
(2) This paper proposes a new image quality restoration framework color space transformation framework (CST-Net). This framework utilizes the transformation of color space. The paper explains why this transformation is used.
(3) The dataset proposed in the paper uses a reliable night rain streak imaging model.
Weaknesses：
(1) The use of image restoration technology based on color space transformation is not a very novel strategy.
(2) The deraining network proposed in the paper is mainly introduced in the supplementary materials. The network architecture and modules used are relatively common. For example, Squeeze-Excitation Residual Block (SERB) and so on.
(3) The loss function used in the paper is also relatively common. This makes the paper lack necessary innovations in network design and training process.

---

> ### Author Rebuttal · Authors · 2025-07-31
>
> We thank Reviewer bmcM for his\her efforts. We are encouraged by the reviewer’s positive comments on the dataset contribution and very helpful research. We answer the questions below and will incorporate all feedback in the revised version.
>
> **1. Novelty on color space transformation**
>
> Our novelty in color space transformation lies in introducing a **learnable nonlinear** coefficient matrix to replace the traditional **fixed linear** transformation used in previous methods (as in the reviewer-mentioned paper [1], which describes “using a linear transformation defined by standard conversion matrices”). Specifically, as shown in **Equations (8) and (9) of the main paper**, each transformation weight is replaced with a learnable variable and further processed through a multilayer perceptron (MLP) for nonlinear transformation. This approach not only retains the ability to handle luminance during the color space conversion but also enables adaptive parameter adjustment based on specific datasets and application scenarios. It adapts to various lighting conditions, scene types, and content characteristics, making the extracted features more robust to the complexity and randomness of nighttime rain effects. We will add a discussion with [1] in the revised paper.
>
> Ref:
>
> [1] Fang, Wenxuan, et al. “Guided real image dehazing using YCbCr color space.” AAAI 2025.
>
> **2. Innovations in network design and training process**
>
> We would like to claim that the main innovation of our method lies in formulating **a new end-to-end trainable two-stage framework** for better nighttime image deraining. We decompose the complex nighttime deraining task into a degradation removal stage and a color refinement stage for joint learning. Although we use some common modules and loss functions as the base backbone, our focus lies in developing **two new components** to better help the feature learning of these two stages. Firstly, we develop a **learnable Color Space Converter (CSC)** in the degradation removal stage to better facilitate nighttime rain removal in the Y channel. Second, instead of employing explicit illumination models for brightness estimation as in existing methods, we introduce a new **Implicit Illumination Guidance (IIG)** module in the color refinement stage, which leverages implicit neural representation technology to encode complex illumination information. We will clarify this in the revised paper.
>
> **3. Issues of t-SNE distribution**
>
> The t-SNE distribution figure shows that our dataset more closely aligns with the distribution of real-world nighttime rainy images. To prove this and avoid the influence of different background content on the results, as stated in the caption of Figure 3 in the main paper, we select samples with similar nighttime street backgrounds and use ResNet50 to extract features. After applying t-SNE dimensionality reduction, images from the same dataset exhibit clustering due to the similarity in rain streak features. By doing so, the distance between different data distributions more accurately represents the size of their rain distribution differences. Note that, due to the domain gap between the synthetic dataset and real-world data, there is low overlap between the different distribution clusters.
>
> In this paper, our goal is to enhance the quality and realism of the synthetic nighttime rain images to reduce the domain gap with real-world data. As shown in the t-SNE distribution visualization, compared to other existing datasets, our proposed dataset successfully narrows the domain gap with real images, which further indicates that our dataset synthesis pipeline generates nighttime rain patterns that are closer to real-world conditions.

---

> ### Comment · Reviewer_bmcM · 2025-08-06
> **Thanks for the response!**
>
> Thank you very much! The author's reply was very detailed and addressed my main concerns. The three references provided by the reviewer regarding the application of color space in rain removal further helped me understand the paper. Best of luck with the paper!

---

> > ### Author Response · Authors · 2025-08-06
> >
> > Thank you for your acknowledgment of our work and responses. We appreciate your constructive feedback that has helped refine our research. Please feel free to reach out if you have further queries or need additional clarification on our work.

---

### Official Review · Reviewer_aW1g · 2025-07-01

**Clarity:** 3
**Significance:** 3
**Originality:** 3
**Rating:** 4
**Confidence:** 3

**Summary:**

This work rethinks nighttime image deraining by addressing key issues in both data and methodology. The authors first introduce HQ-NightRain, a new high-quality dataset that generates more realistic rainy scenes by modeling the dependency of rain visibility on local illumination. To leverage this, they also propose CST-Net, a novel framework that uses a learnable color space converter to perform deraining in the more effective Y (luminance) channel. The proposed method achieves state-of-the-art results on several synthetic and real-world benchmarks.

**Questions:**

1. The data synthesis pipeline uses a 3x3 convolution, f[·], to merge the background and rain masks. Could you provide more justification for this specific choice over simpler methods like linear addition or alternative network structures? It would be valuable to understand if an ablation study was performed to assess the sensitivity of the final dataset's quality to this merge strategy.
2. The paper effectively demonstrates the proposed dataset's superiority through generalization experiments (Table 3), which evaluates the synthesis pipeline holistically. Was any analysis done to isolate the contributions of individual components within the pipeline? For instance, what is the quantitative impact of the illumination-aware blending (σ(·)) versus the defocus blur (ρ(·)) on the final model's performance?
3. The architecture's central premise—that rain artifacts are overwhelmingly present in the Y-channel—is challenged by scenarios involving colored rain streaks. Could you comment on how the model performs in such cases? Specifically, does passing the noisy Cb/Cr channels directly to the refinement stage compromise its ability to perform accurate color correction and potentially lead to color artifacts in the final output?

**Ethical Concerns:**

["NO or VERY MINOR ethics concerns only"]

**Final Justification:**

Based on the author's response and the comments of other reviewers, I gave the final score.

**Limitations:**

1. The dataset synthesis pipeline, while innovative, relies on certain simplifications. It uses fixed illumination thresholds, which may not be optimal for all lighting scenarios, and models complex raindrop optics solely with defocus blur.
2. The work's scope is focused narrowly on deraining and does not address the common real-world challenge of mixed adverse conditions (e.g., rain with fog), where the robustness of the Y-channel-centric approach remains unevaluated.

**Paper Formatting Concerns:**

No significant formatting issues were observed.

**Quality:**

2

**Strengths And Weaknesses:**

Strength：
Dataset Contribution: The introduction of the HQ-NightRain dataset is a significant contribution, featuring an illumination-aware synthesis pipeline that addresses a key limitation in prior work by generating more physically-grounded, non-uniform rain.
Model Framework Innovation:The methodological novelty of the CST-Net framework centers on its learnable color space converter (CSC), which moves beyond fixed transformations to adaptively learn an optimal feature space for the deraining task.
Excellent Performance: The proposed method demonstrates state-of-the-art (SOTA) performance on multiple synthetic and real-world benchmarks.
Weaknesses：
Reliance on Fixed Hyperparameters: The dataset construction relies on fixed illumination thresholds. This one-size-fits-all approach may be sub-optimal for synthesizing images from backgrounds with diverse lighting conditions (e.g., a dimly lit street vs. a bright downtown area), potentially limiting the dataset's robustness and generalization.
Simplifications in the Synthesis Pipeline: While the illumination-aware concept is novel, the synthesis pipeline involves certain simplifications. The use of fixed illumination thresholds and the modeling of raindrops solely through defocus blur are approximations that do not fully capture the complexity of real-world optics, suggesting that the dataset's realism could be further enhanced.
Limited Scope to Single Degradation: The work's focus is narrowly on deraining and does not address the common challenge of mixed adverse conditions. Real-world nighttime scenes often feature rain combined with fog or haze, and the paper provides no analysis of how robust the proposed method—particularly its core reliance on the Y-channel—would be in these more complex, multi-degradation scenarios.

---

> ### Author Rebuttal · Authors · 2025-07-31
>
> We thank Reviewer aW1g for his\her efforts. We are encouraged by the reviewer’s positive comments on our dataset contribution, model framework innovation, and excellent performance. We answer the questions below and will incorporate all feedback in the revised version.
>
> **1. Reliance on fixed hyperparameters**
>
> In our dataset construction, our thresholding strategy is based on observations from a large number of real nighttime rainy images, where rain streaks are nearly imperceptible in extremely dark or overly bright regions. To justify our choice of hyperparameters, we have provided a user study and validation results in **Sections H and L of the supplementary material**, respectively. As shown in **Table 11 of the supplementary material**, we tested random thresholds as well as various threshold combinations. The results demonstrate that our chosen thresholds yield the best performance. To further assess the robustness and generalization capability of different threshold settings, we trained models on datasets generated with different threshold values and evaluated them on the real-world RealRain1k-L dataset. Our fixed threshold setting achieved the best performance, supporting its effectiveness and rationality.
>
> Furthermore, as illustrated in **Figure 15 of the supplementary material**, we have conducted an online user study to evaluate the perceptual quality of images generated with different threshold settings. The results show that our setting received the highest preference, indicating that our dataset is more consistent with human perception and better aligned with real-world visual appearance.
>
> **2. Simplifications in the synthesis pipeline**
>
> The core of our data synthesis pipeline lies in the incorporation of illumination information from background images. The resulting non-uniform rain masks align with nighttime imaging characteristics, significantly enhancing the realism of the dataset compared to existing synthesis methods. As shown in **Figure 16 of the supplementary material**, user study results indicate a competitive preference for our synthesized images, suggesting better alignment with human perception. In future work, we will continue to explore advanced synthesis strategies to better reflect real-world optical complexities and further improve the realism of our dataset.
>
> **3. Limited scope to single degradation**
>
> We would like to claim that our method not only solves the deraining task but also handles mixed adverse conditions. As shown in **Table 6 of the main paper**, we have provided experimental results on the Multi-Weather6k dataset, which includes rain, snow, and haze scenarios. The results indicate that our model has promising potential for restoring images degraded by mixed adverse weather conditions.
>
> As for the robustness in more complex multi-degradation scenarios, we have evaluated on the RealRain1k dataset, which includes both daytime and nighttime rainy scenes, and the RainDS-real dataset, which contains rain streaks, raindrops, and their combinations, as shown in **Table 2 of the main paper.** The results demonstrate that our method achieves robust performance across these diverse real-world conditions. Qualitative results are presented in **Figures 24, 25, and 26 of the supplementary material**.
>
> The effectiveness of our model in addressing the above issues stems from its core design: a learnable mapping from the RGB color space to a luminance-dominated domain, which enables more effective degradation modeling. Since degradations such as snowfall also significantly affect image brightness and contrast, their visual artifacts become more prominent in the Y channel. As a result, CST-Net is not limited to nighttime rain degradation but can also handle other degradation types that impact luminance structures. In addition, our proposed Implicit Illumination Guidance (IIG) helps disentangle illumination and structure at the feature level, allowing the model to capture degradation-specific patterns under diverse conditions and guide the restoration process accordingly. Finally, our training paradigm does not rely on degradation-specific structural priors, thus demonstrating strong transferability across various weather-related degradation scenarios.
>
> For rain combined with fog or haze, we have discussed the limitations in **Section 6 of the main paper**. In future work, we will further consider expanding the extra dataset under low-light conditions to include veiling effects and explore the incorporation of physical models to address this problem.
>
> **4. Issues of the data synthesis pipeline**
>
> The traditional linear addition method adopts Equation (1) in the main paper, $R_s=B+S$, where rain streaks $S$ are simply added to the clean background $B$. This simpler addition ignores the interaction between rain streaks and the background, often resulting in a noticeable “sticker-like” effect. This is particularly evident around object boundaries, where rain streaks may unnaturally overlap with edges. As shown in the example image in **Figure 3(a) of the main paper.**
>
> In our paper, we adopt Equation (2) in the main paper, $R_s=f[B,\sigma(S)]$, where a 3×3 convolution operation $f[\cdot]$ is applied to model the local interactions between rain streaks and the background. Rather than treating rain as pixel-wise additive noise, this approach introduces contextual perturbations to surrounding pixels, more accurately reflecting the physical process of rain formation in real-world imagery.
>
> We further compare the impact of different fusion strategies on the final dataset's quality, and the results are shown in the table below. The quantitative results show that our merge strategy achieves the best performance in the NIQE [1] and NRQM [2] non-reference metrics, significantly improving the quality of the dataset. We will add these results in the revised paper.
>
> | Methods | Linear addition | Convolutional  merge $f[\cdot]$ | NIQE⬇ | NRQM⬆ |
> | :-----: | :-------------: | :-----------------------------: | :---: | :---: |
> |   M1    |        √        |                                 | 5.93  | 5.45  |
> |  Ours   |                 |                √                | 5.28  | 6.37  |
>
>
> Ref:
> [1] Anish Mittal, et al. “Making a completely blind image quality analyzer.” SPL 2012.
> [2] Ma, Chao, et al. “Learning a no-reference quality metric for single-image super-resolution.” CVIU 2017.
>
> **5. Effect of data synthesis pipeline components on model performance**
>
> According to the reviewer’s suggestion, we further add experiments to evaluate the effect of each component in the data synthesis pipeline on the final model’s performance. Specifically, we analyze our model performance on datasets generated using different synthesis pipeline configurations, and the results are shown in the table below. As observed, our data synthesis pipeline incorporates convolutional merge $f[\cdot]$, illumination merge $\sigma[\cdot]$, and defocus blur $\rho[\cdot]$, which better reflect real-world degradation conditions. We will add these results in the revised paper.
>
> | Methods | Linear  addition | Convolutional  merge $f[\cdot]$ | Illumination  merge $\sigma[\cdot]$ | Defocus  blur $\rho[\cdot]$ | PSNR ⬆  | SSIM  ⬆ |
> | :-----: | :--------------: | :-----------------------------: | :---------------------------------: | :-------------------------: | :-----: | :-----: |
> |   D1    |        √         |                                 |                                     |                             | 25.3894 | 0.8705  |
> |   D2    |                  |                √                |                                     |                             | 28.5863 | 0.9456  |
> |   D3    |                  |                √                |                  √                  |                             | 36.5540 | 0.9754  |
> |  Ours   |                  |                √                |                  √                  |              √              | 31.9161 | 0.9493  |
>
>
> **6. How the model performs in colored rain streak removal?**
>
> As shown in **Figure 22 of the supplementary material**, our method effectively removes colored rain streaks influenced by yellow light sources, demonstrating strong robustness in handling colored rain streaks. If the Cb/Cr channel features are directly passed to the color refinement stage, it would not cause color artifacts. The underlying reason is that we designed the Implicit Illumination Guidance (IIG) module, which guides the model’s attention toward complex rain streaks near light sources (including colored rain streaks) and extracts implicit degradation information to guide the second-stage color refinement process.

---

> > ### Comment · Reviewer_aW1g · 2025-08-06
> >
> > Thanks for the authors' response. It has addressed the majority of my concerns. I will maintain my original score.

---

### Official Review · Reviewer_jK5Y · 2025-07-02

**Clarity:** 4
**Significance:** 3
**Originality:** 3
**Rating:** 5
**Confidence:** 4

**Summary:**

This paper introduces a high-quality benchmark dataset, **HQ-NightRain**, for nighttime image deraining, enhancing the realism of synthetic images. It proposes a robust, learnable **color space transformation** framework that leverages the **Y channel** for improved rain removal. Experimental results show that the method outperforms existing approaches, demonstrating the effectiveness of both the dataset and the proposed model.

**Questions:**

### Major issues
1. About Fig. 1.
    - Are the histogram statistics based on real or synthetic nighttime rainy images?
    - Is the histogram computed from a single image pair (with and without rain)? If so, please provide the corresponding image. If it is based on multiple images (as expected for a histogram), please show examples of both rainy and clean images. Since the Y channel mainly reflects brightness, the results could be influenced by the lighting conditions in nighttime scenes.


### Minor issues
1. On page 2, line 42, the paper states that existing methods remove rain in the RGB color space rather than the YCbCr space. However, some prior works have already explored deraining in the YCbCr color space as shown in their publicly available code. Please cite them [1-3] appropriately.

### References

[1] Wei, Wei, et al. "Should we encode rain streaks in video as deterministic or stochastic?." Proceedings of the IEEE International Conference on Computer Vision. 2017.

[2] Li, Minghan, et al. "Video rain streak removal by multiscale convolutional sparse coding." Proceedings of the IEEE conference on computer vision and pattern recognition. 2018.

[3] Li, Minghan, et al. "Online rain/snow removal from surveillance videos." IEEE Transactions on Image Processing 30 (2021): 2029-2044.

**Ethical Concerns:**

["NO or VERY MINOR ethics concerns only"]

**Final Justification:**

Taking into account the other reviewers’ comments and the authors’ detailed rebuttal, I believe the paper makes a valuable contribution and recommend acceptance.

**Limitations:**

***Future work:*** This paper demonstrates the effectiveness of CST-Net for image deraining by operating in the Y channel. Can this approach be extended to video deraining tasks? Is it also adaptable to snow removal?

**Paper Formatting Concerns:**

No formatting concerns

**Quality:**

4

**Strengths And Weaknesses:**

### Strengths
1. HQ-NightRain considers the visibility of rain under varying illumination conditions by using illumination coefficient matrix. This design reduces the domain gap between synthetic and real-world rainy videos.

2. This paper proposes CST-Net, an effective color space transformation framework CST-Net for nighttime image deraining.

3. The paper is well written. Figures 1–4 and all equations clearly illustrate the main motivation and contributions.

4. Experiments are comprehensive and results are solid. This paper provides experiemntal results on multiple benchmarks, including HQ-NightRain, public and real-world datasets.

### Weaknesses
1. Future work. This paper demonstrates the effectiveness of CST-Net for image deraining by operating in the Y channel. Can this approach be extended to video deraining tasks? Is it also adaptable to snow removal?

---

> ### Author Rebuttal · Authors · 2025-07-31
>
> We thank Reviewer jK5Y for his\her efforts. We are encouraged by the reviewer’s positive comments on the dataset quality, effective method, paper presentation, and solid experiments. We answer the questions below and will incorporate all feedback in the revised version.
>
>
>
>  **1. Extend to video deraining and snow removal**
>
> For snow removal task, we have provided the experimental results on the multi-weather dataset Multi-Weather6k (including rain, snow, and haze) in **Table 6 of the main paper** and elaborate on them in **Section 6.5**, demonstrating that our method is also effective for image desnowing.
>
>
>
> For the video deraining task, we use the RainVID&SS [1] dataset, where the Cam-Vid subset consists of training frames extracted from 3 long clips and testing frames extracted from 2 long clips. We adopt this subset and train all models using patches of size 128×128 pixels. We compare our method with two video deraining approaches, S2VD [2] and MPEVNet [1], and the quantitative results are shown in the table below. The results demonstrate that our CST-Net still exhibits strong restoration potential in the video deraining task. We will add these results in the revised paper.
>
>
>
> | Methods |  S2VD   | MPEVNet | CST-Net (Ours) |
> | :-----: | :-----: | :-----: | :------------: |
> | PSNR ⬆  | 27.6059 | 31.3213 |    31.8259     |
> | SSIM ⬆  | 0.8755  | 0.9252  |     0.9366     |
>
>
>
> Ref:
>
> [1] Sun, Shangquan, et al. “Event-aware video deraining via multi-patch progressive learning.” IEEE TIP 2023.
>
> [2] Yue, Zongsheng, et al. “Semi-supervised video deraining with dynamical rain generator.” CVPR 2021.
>
>
>
>  **2. Issues of Fig. 1**
>
> The histogram statistics in Fig. 1 are based on multiple paired images from the public RealRain1k dataset. Due to rebuttal format limitations, we are unable to provide corresponding image pair examples here. Note that we normalize the background luminance before analyzing the rain effects in the Y channel. By doing so, our approach ensures that the analysis focuses on the nighttime rain features themselves, rather than being dominated by lighting conditions in the scene. We will clarify this in the revised paper.
>
>
>
>  **3. Missing references**
>
> Thanks for the suggestion. We will add the suggested references in the revised paper.

---

> > ### Comment · Reviewer_jK5Y · 2025-08-05
> >
> > Thanks for your rebuttal, it addressed my concerns. Best of luck with the paper!

---

> > > ### Author Response · Authors · 2025-08-06
> > >
> > > Thank you for your acknowledgment of our work and responses. We appreciate your constructive feedback that has helped refine our research. Please feel free to reach out if you have further queries or need additional clarification on our work.

---

### Decision · Program_Chairs · 2025-09-17

**Decision:**

Accept (poster)

**Comment:**

This paper introduces a new benchmark dataset (HQ-NightRain) and a learnable color space transformation framework (CST-Net) for nighttime image deraining. Reviewers agree that the paper is technically solid, clearly presented, and supported by comprehensive experiments. The dataset is a timely and valuable contribution, and the proposed method demonstrates strong effectiveness and robustness across multiple benchmarks. While some reviewers raised concerns about novelty and simplifications in dataset synthesis, the authors provided thorough rebuttals and additional results that addressed these issues satisfactorily. Overall, the strengths outweigh the weaknesses, and all reviewers support acceptance.